# Patched1 and Patched2 inhibit Smoothened non-cell autonomously

**Brock Roberts, Catalina Casillas, Astrid C Alfaro, Carina Jägers, Henk Roelink\***

Department of Molecular and Cell Biology, University of California, Berkeley, Berkeley, United States

**Abstract** Smoothened (Smo) inhibition by Patched (Ptch) is central to Hedgehog (Hh) signaling. Ptch, a proton driven antiporter, is required for Smo inhibition via an unknown mechanism. Hh ligand binding to Ptch reverses this inhibition and activated Smo initiates the Hh response. To determine whether Ptch inhibits Smo strictly in the same cell or also mediates non-cell-autonomous Smo inhibition, we generated genetically mosaic neuralized embryoid bodies (nEBs) from mouse embryonic stem cells (mESCs). These experiments utilized novel mESC lines in which *Ptch1, Ptch2, Smo, Shh* and *7dhcr* were inactivated via gene editing in multiple combinations, allowing us to measure non-cell autonomous interactions between cells with differing Ptch1/2 status. In several independent assays, the Hh response was repressed by Ptch1/2 in nearby cells. When *7dhcr* was targeted, cells displayed elevated non-cell autonomous inhibition. These findings support a model in which Ptch1/2 mediate secretion of a Smo-inhibitory cholesterol precursor.

## Introduction

Hedgehog (Hh) signaling is critically important during embryonic development and its aberrant regulation is associated with common, lethal birth defects and cancers. Conserved roles as a morphogen and in tissue homeostasis make Hh signaling fundamental to most forms of metazoan life (*Briscoe and Thérond, 2013*; *Hooper and Scott, 2005*; *Ingham and McMahon, 2001*).

Smoothened (Smo) and Patched (Ptch; Ptch1 and Ptch2 in amniotes) are conserved multipass transmembrane proteins required for proper Hh pathway transduction. Smoothened is a putative G-protein-coupled receptor and Ptch has homology to a family of proton-driven antiporters. The regulatory relationship between Ptch and Smo has been the subject of much study, resulting in the following model: (1) Ptch in its unbound state inhibits Smo cell autonomously (2) Hh ligand bound to Ptch releases this inhibition and (3) uninhibited Smo redistributes in the cell and activates transcription of target genes through downstream factors.

While this model has wide acceptance, the Ptch-dependent mechanism responsible for Smo repression has proven elusive. Ptch belongs to the Resistance, Nodulation and Division (RND) family of proton-driven, trimeric efflux pumps that are ubiquitously present in all studied organisms (*Nikaido and Takatsuka, 2009*). RNDs secrete diverse molecular cargos, including lipophilic and amphiphilic molecules such as antibiotics and sterols. They are well studied in Gram-negative bacteria, where they confer multidrug resistance via antibiotic efflux (*Tseng et al., 1999*).

According to prevailing models, Ptch inhibits Smo sub-stoichiometrically rather than through a direct binding relationship, by regulating the localization of a Smo regulatory molecule (*Taipale et al., 2002*). Despite the discovery of exogenous and endogenous molecules capable of regulating Smo, no Smo-regulatory Ptch substrate has been identified (*Sharpe et al., 2015*). Nevertheless, several observations indicate that the endogenous cargo of Ptch is a steroidal molecule: (1) the plant-derived steroidal alkaloid cyclopamine binds Smo and inhibits the Hh response (*Chen et al., 2002a*; *Incardona et al., 1998*); (2) heterologous Ptch expression in yeast enhances

*For correspondence: roelink@berkeley.edu

**Competing interests:** The authors declare that no competing interests exist.

BODIPY-cholesterol efflux (*Bidet et al., 2011*); (3) the closest prokaryotic homolog of Ptch, HpnH, transports bacterial sterols (hopanoids) from the inner to the outer bacterial membrane (*Doughty et al., 2011*); (4) 7-dehydroxycholesterol reductase (7DHCR), catalyzes the conversion of 7DHC into cholesterol and genetic loss of *7dhcr* is associated with defects in Shh signaling, perhaps via accumulation of a late sterol precursor (or its derivative) that inhibits Smo (*Bijlsma et al., 2006*; *Cohen, 2010*; *Gruchy et al., 2014*; *Incardona et al., 2000a*; *Linder et al., 2015*; *Sever et al., 2016*); (5) Ptch has a sterol-sensing domain (SSD) that is conserved within sterol biogenesis regulatory enzymes, and thus likely binds sterols (*Incardona, 2005*), and this domain is necessary for Smo inhibition by Ptch in *Drosophila* (*Strutt et al., 2001*). Within the third transmembrane domain of the SSD (the fourth transmembrane domain of Ptch1) resides a universally conserved Aspartic acid residue that when mutated in bacterial RNDs blocks transport (*Zgurskaya and Nikaido, 1999*). Mutation of this residue in Ptch1 yields an allele unable to inhibit Smo both in vivo and in vitro (*Alfaro et al., 2014*; *Strutt et al., 2001*; *Taipale et al., 2000*). These observations have led to the hypothesis that Ptch1/2 re-localizes a cholesterol precursor that is inhibitory to Smo (*Incardona et al., 1998*).

As a proton-driven antiporter of the RND family, Ptch1/2 is predicted to secrete its cargo. The observation that murine fibroblasts overexpressing Ptch1 can condition their supernatant with a Smo inhibitor supports this notion (*Bijlsma et al., 2006*). However, few reports address non-cell-autonomous Smo regulation by Ptch1 antiporter activity. This may be due to other non-cell autonomous mechanisms of Ptch-mediated inhibition unrelated to its antiporter activity, such as its proposed ability to sequester Hedgehog ligands from the environment and thus suppress the Hh response (*Chen and Struhl, 1996*; *Incardona et al., 2000b*; *Milenkovic et al., 1999*; *Strutt et al., 2001*). Ligand sequestration by Ptch thus complicates efforts to assess non-cell autonomous antiporter-mediated Ptch activity. Besides these possible non-cell autonomous activities, Ptch plays a cell autonomous role in the activation of Smo via the accumulation of phosphatidylinositol 4-phosphate (*Jiang et al., 2016*; *Yavari et al., 2010*) that can activate Smo via its intracellular C-terminal domain.

We attempted to address the non-cell autonomous contribution of Ptch1/2 to Smo regulation with genetically mosaic neural tissue derived from genome-edited mouse embryonic stem cells (mESCs). As a morphogen, Sonic Hedgehog (Shh) patterns the embryonic vertebrate neural tube through a well-studied transcriptional response (*Cohen et al., 2013*; *Roelink et al., 1994*). Shh is expressed ventrally in embryos in the notochord and floor plate, yielding a ventral to dorsal gradient of Hh pathway activity in which ventral cell types have a high level of pathway activation. We can effectively model these signaling events in vitro by differentiating genetically distinct stem cells into neuralized embryoid bodies (nEBs) (*Meinhardt et al., 2014*; *Wichterle et al., 2002*). nEBs have previously been shown to be highly responsive to Shh, the Smo agonist SAG, and cyclopamine, indicating that Smo activity is subject to regulation in this system (*Frank-Kamenetsky et al., 2002*). We have also found that Smo becomes maximally activated in nEBs lacking Ptch1 and Ptch2 (*Alfaro et al., 2014*).

In our experimental approach, cells in one compartment of genetically mosaic nEBs are either proficient or genetically null for *Ptch1/2*. We measured Hh pathway activity, and thus assess Ptch1/2-mediated non-cell autonomous Smo inhibition in a separate mosaic compartment designed to have active Smo. If the null hypothesis of Ptch1/2 as strict cell-autonomous Smo inhibitors is true, we predict that in mosaic tissues in which cell differ in regard to their Ptch1/2 status the resulting level of Hh response is the average of both constituent cells cultured alone. A rejected null hypothesis supports the notion that Ptch1/2 can inhibit Smo activity non-cell autonomously.

Using genome editing with Tal endonucleases (TALENs) and CRISPR/Cas9, we generated mESC lines genetically null for *Ptch1, Ptch2, Smo, Shh* and *7dhcr* in many combinations. We show that each cell line differentiates as monotypic nEBs to neural progenitor fates predicted according to the established Hh signaling model. We then demonstrate that within genetically mosaic nEBs, cells with Ptch1/2 activity inhibit the Hh response non-cell autonomously in neighboring cells deficient for Ptch1/2 that contain activated Smo. Ptch1/2 also inhibits the response of neighboring wild-type cells to Shh and the Smo agonist SAG. Loss of 7DHCR activity results in an increased ability of Ptch1/2 proficient cells to inhibit the Hh response non-cell autonomously. We attribute these observations to a fundamental function of Ptch1/2 in secreting a steroidal Smo inhibitor via its proton antiporter activity.

## Results

### Ptch1/2 activity inhibits Smo both cell autonomously and non-cell autonomously

In order to assess if Ptch1/2 activity inhibits Smo in neighboring cells, we established a panel of genome-edited mESC lines harboring null mutations in the Hh pathway genes *Ptch1*, *Ptch2*, *Smo* and *Shh*. We then co-cultured these cell lines in genetically mosaic nEBs. We used pre-existing mutant cell lines and TAL effector endonucleases (TALENs) to generate our initial mESC panel (*Cermak et al., 2011*). This approach presents an in vitro model in which we can measure the non-cell autonomous effects of Ptch1/2 by varying the Ptch1/2 status of cells and measuring the effect on the Hh response in a specific subset of neighboring cells.

Before using these cell lines in genetically mosaic experiments, we first confirmed that nEBs derived from each cell line in our panel differentiated as expected, given their *Ptch1*, *Ptch2*, *Shh* and *Smo* genotype. We predicted that upon neural differentiation each cell line would acquire a neural progenitor identity reflecting the status of its core Hh pathway regulatory genes. Immunostaining for four markers of distinct neural progenitor populations along the vertebrate dorsoventral axis was quantified in order to assess identity. Nkx2.2, Olig2 and Isl1/2 served as markers of ventral cell populations with high Hh activity, while Pax7 designated dorsal tissue where the pathway is silent.

The prevailing model for Hh signaling in the neural tube guided our predictions for each cell line. For example, *Ptch1*$^{+/LacZ}$;*Shh*$^{-/-}$ mESCs yielded nEBs with little Hh pathway activity and thus high dorsal identity, as indicated by a gain of dorsal Pax7$^+$ cells to levels greater than wild type (*Figure 1A, B,J,K*). Because *Shh* encodes an activating factor, and Pax7$^+$ cells indicate pathway quiescence, this cell line differentiated in the manner predicted. *Ptch1*$^{LacZ/LacZ}$ nEBs by contrast were highly ventral (*Figure 1D,M*), corroborating previous reports using this cell line, and supporting the canonical view that Ptch1 is a negative pathway regulator (*Ptch1*$^{LacZ}$ is an established null allele [*Goodrich et al., 1997*; *Rohatgi et al., 2007*]). Shh signaling through Ptch2 in cells null for *Ptch1* has also been observed and thus unsurprisingly *Ptch1*$^{LacZ/LacZ}$;*Shh*$^{-/-}$ nEBs had reduced numbers of Nkx2.2$^+$/Olig2$^+$ cells (*Figure 1E,N*), compared to *Ptch1*$^{LacZ/LacZ}$ nEBs, indicating ligand dependency of the response. As previously reported, *Ptch1*$^{LacZ/LacZ}$;*Ptch2*$^{-/-}$ nEBs differentiated into identities associated with high Hh pathway activity, indicated by robust Nkx2.2, Isl1/2 and Olig2 expression (*Figure 1G,P*) (*Alfaro et al., 2014*). *Ptch1*$^{LacZ/LacZ}$;*Ptch2*$^{-/-}$;*Shh*$^{-/-}$ nEBs had unaffected ventral identity (*Figure 1H, Q*), consistent with a model for Shh-independent Hh pathway activation in the dual absence of Ptch1/2. Smo was invariably required for the activation of the Hh pathway, as established models predict. *Ptch1*$^{LacZ/LacZ}$;*Smo*$^{-/-}$ and *Ptch1*$^{LacZ/LacZ}$;*Ptch2*$^{-/-}$;*Smo*$^{-/-}$ nEBs were entirely lacking Nkx2.2, Isl1/2 and Olig2 expression and instead expressed Pax7, as did *Smo*$^{-/-}$ nEBs (*Figure 1C,F,I,L,O,R*). All *Smo*$^{-/-}$ nEBs thus conformed to the standard signaling model by acquiring highly dorsal fates as cells refractory to Shh pathway activation. Each clone had abundant Pax6$^+$ nuclei and Tuj1 expression, suggesting robust neuralization (*Figure 1—figure supplement 1*).

Because of their high level of Hh pathway activity, we used *Ptch1*$^{LacZ/LacZ}$;*Ptch2*$^{-/-}$;*Shh*$^{-/-}$ mESCs as sensitized cells in which non-cell autonomous Smo-inhibitory effects mediated by Ptch1/2 in adjacent cells could be measured. We reasoned that genetic ablation of both Ptch1 and Ptch2 would be necessary to assess non-cell autonomous Smo regulation in these cells because Ptch2 compensates for Ptch1 loss in cell autonomous Smo regulation (*Alfaro et al., 2014*; *Zhulyn et al., 2015*). We previously reported that the Hh pathway in *Ptch1*$^{LacZ/LacZ}$;*Ptch2*$^{-/-}$ nEBs could not be further activated by the Smo agonist SAG (*Chen et al., 2002b*), suggesting maximal Smo activation in the absence of Ptch1/2. To assess ligand-independent cell non-autonomous signaling, *Shh*$^{-/-}$ cells were necessary because Shh is expressed in *Ptch1*$^{-/-}$; *Ptch2*$^{-/-}$ nEBs and the *Ptch1*$^{-/-}$ mouse neural tube (*Alfaro et al., 2014*; *Goodrich et al., 1997*).

We validated that *Ptch1*$^{LacZ/LacZ}$;*Ptch2*$^{-/-}$;*Shh*$^{-/-}$ cells differentiate into ventral neural fates in a Smo-dependent, Shh-independent manner. First, we treated nEBs with the Smo inhibitor cyclopamine (*Cooper et al., 1998*; *Gaffield et al., 1999*; *Incardona et al., 2000a*) and found that cyclopamine inhibited Isl1/2 and Nkx2.2 expression with an IC$_{50}$ around 25 nM (*Figure 1S*). We found that ShhN (soluble N-terminal Shh) was able to induce the Hh response both in *Ptch1*$^{+/LacZ}$;*Shh*$^{-/-}$ and *Ptch1*$^{LacZ/LacZ}$;*Shh*$^{-/-}$ nEBs but not in *Ptch1*$^{LacZ/LacZ}$;*Ptch2*$^{-/-}$;*Shh*$^{-/-}$ nEBs, (*Figure 1T*), corroborating our report that *Ptch1*$^{-/-}$ cells respond to Shh via Ptch2 (*Alfaro et al., 2014*). To test if the response to Shh can be restored in *Ptch1*$^{LacZ/LacZ}$;*Ptch2*$^{-/-}$ cells by Ptch1 expression, we derived a fibroblast-like

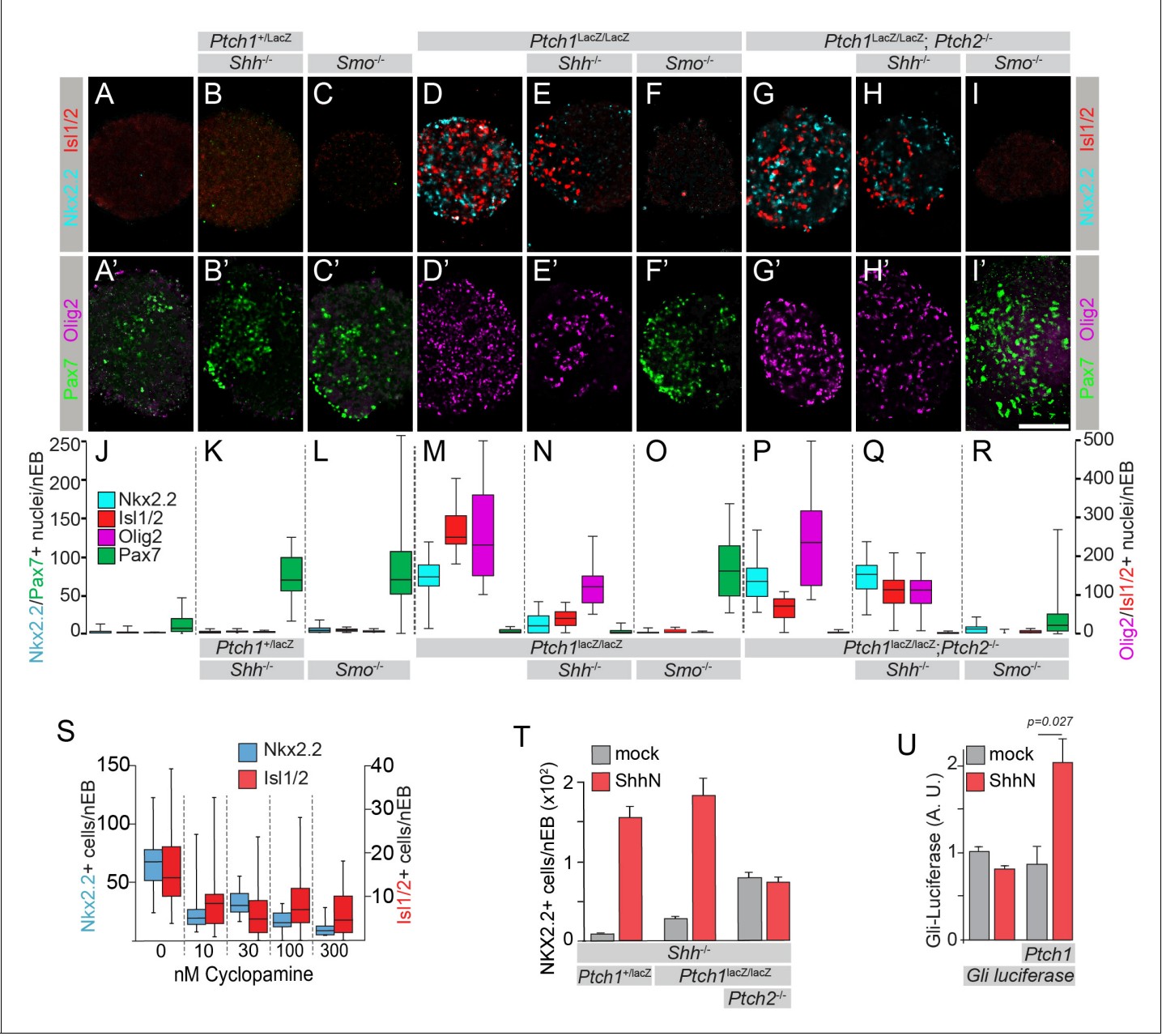

**Figure 1.** Ptch1/2 disruption enhances Smo-dependent activation of the Hh response. (A–I') Genome-edited mESCs with various genotypes were differentiated into nEBs and stained for ventral markers Nkx2.2 (Cyan), Olig2 (Magenta) and Isl1/2 (Red), indicating Hh pathway activation, and the dorsal marker Pax7 (Green) indicating Hh pathway inactivity. (J–R) Nkx2.2, Isl1/2, Olig2 or Pax7 staining quantification (box and whiskers). WTnEBs have a quiescent Hh response (A,A',J). Loss of Shh (Ptch1$^{+/L}$ (B,B',K) or Smo (Smo$^{-/-}$) (C,C',L) in cells with intact Ptch1/2 function (Ptch1$^{LacZ}$ is null) increases Pax7. nEBs without Ptch1 (Ptch1$^{LacZ/LacZ}$, D,D',M) have an activated Hh response. Loss of Shh (Ptch1$^{LacZ/LacZ}$;Shh$^{-/-}$, E,E',N) decreases the response. Ptch1$^{LacZ/LacZ}$;Smo$^{-/-}$ nEBs acquire Pax7 (F,F',O) and are devoid of all three ventral markers. nEBs lacking all Ptch1/2 activity (Ptch1$^{LacZ/LacZ}$;Ptch2$^{-/-}$) have an activated Hh response pathway (G,G',P). Ptch1$^{LacZ/LacZ}$;Ptch2$^{-/-}$;Shh$^{-/-}$ nEBs retain similar ventral identity (H,H',Q). Ptch1$^{LacZ/LacZ}$;Ptch2$^{-/-}$;Smo$^{-/-}$ nEBs lose ventral identity and express Pax7 (I,I',R). Scale bar is 100 μm. (S) Nkx2.2 and Isl1/2 expression in Ptch1$^{LacZ/LacZ}$;Ptch2$^{-/-}$ nEBs is quantified (box and whisker) in the presence of 0–300 nM cyclopamine. (T) Ptch1$^{+/LacZ}$;Shh$^{-/-}$ and Ptch1$^{LacZ/LacZ}$;Shh$^{-/-}$, but not Ptch1$^{LacZ/LacZ}$;Ptch2$^{-/-}$;Shh$^{-/-}$ retain their ability to respond to exogenously supplied ShhN by inducing Nkx2.2 expression. (U) Fibroblast-like cells derived from Ptch1$^{LacZ/LacZ}$;Ptch2$^{-/-}$ mESCs were transfected with a Gli:luciferase construct alone or together with Ptch1. Independently, Ptch1$^{LacZ/LacZ}$;Ptch2$^{-/-}$ cells were mock transfected or transfected with ShhN. Gli:Luciferase is quantified in co-cultures. Ptch1$^{LacZ/LacZ}$;Ptch2$^{-/-}$ cells expressing Ptch1 can respond to ShhN supplied in co-cultured cells. p-value is indicated, n = 6. Variance is s.e.m. in A–U.

The following figure supplement is available for figure 1:

*Figure 1 continued on next page*

Figure 1 continued

**Figure supplement 1.** Neural differentiation proceeds normally in all modified cell lines.

cell line (*Anastassiadis et al., 2010*; *Gökhan et al., 1998*). Despite Ptch1/2 absence, these cells have low Hh pathway activation, resembling 24–48 hr nEBs, allowing us to assess Hh pathway induction (Figure 3C). *Ptch1*$^{LacZ/LacZ}$;*Ptch2*$^{-/-}$ fibroblasts co-transfected with *Gli:Luciferase*, a Hh pathway reporter (*Taipale et al., 2000*) and *Ptch1*, then co-cultured with *ShhN*-transfected cells, activate the Hh response (*Figure 1U*). ShhN alone was unable to activate the Hh pathway in *Ptch1*$^{LacZ/LacZ}$; *Ptch2*$^{-/-}$ cells, consistent with results obtained with nEBs (*Figure 1T*). Thus, the ability to respond to exogenous ShhN is lost in the absence of Ptch1/2, but can be restored with *Ptch1* transfection.

We first assayed non-cell autonomous Ptch1/2 inhibition of Smo-mediated neural differentiation by determining whether *Ptch1*$^{LacZ/LacZ}$;*Ptch2*$^{-/-}$;*Shh*$^{-/-}$ cells have diminished ventral progenitors when co-cultured in mosaic nEBs with *Ptch1*$^{+/LacZ}$;*Shh*$^{-/-}$ cells, thus rejecting the null hypothesis of Ptch1/2 acting strictly cell autonomously (*Figure 2A*). Relative to *Ptch1*$^{LacZ/LacZ}$;*Ptch2*$^{-/-}$;*Shh*$^{-/-}$ nEBs, *Ptch1*$^{+/LacZ}$;*Shh*$^{-/-}$ nEBs are devoid of Nkx2.2$^+$ cells and highly diminished for Olig2$^+$ cells (*Figure 2B*), presumably because Smo is under Ptch1/2-mediated repression. Interestingly, mosaic nEBs comprised 1:1 of the two cell lines resembled *Ptch1*$^{+/LacZ}$;*Shh*$^{-/-}$ nEBs as judged by their near complete absence of Nkx2.2$^+$ cells, and had fewer Olig2$^+$ cells than expected for a 1:1 mosaic. Because Shh is genetically absent from these nEBs and *Ptch1*$^{LacZ/LacZ}$;*Ptch2*$^{-/-}$;*Shh*$^{-/-}$ mESCs are ligand insensitive, we interpreted this result as consistent with extracellular flux of Ptch1/2 substrates from *Ptch1*$^{+/LacZ}$;*Shh*$^{-/-}$ mESCs in which Ptch1/2 are intact, and inconsistent with Shh sequestration from this compartment of cells.

We next assayed the differentiation of *Ptch1*$^{LacZ/LacZ}$;*Ptch2*$^{-/-}$;*Shh*$^{-/-}$ cells when co-cultured 1:1 in mosaic nEBs with either *Smo*$^{-/-}$ or *Ptch1*$^{-/-}$;*Ptch2*$^{-/-}$;*Smo*$^{-/-}$ mESCs. We reasoned that ablating Smo would decouple cell autonomous cell fate decisions from Ptch1/2 status. Because we find that *Smo*$^{-/-}$ nEBs retain dorsal identity (Pax7$^+$ cells) regardless of their Ptch1/2 status (*Figure 1C,L,F,O,I,R*), we expected markers of ventral identity in mosaic nEBs to be lineage restricted to the *Ptch1*$^{LacZ/LacZ}$; *Ptch2*$^{-/-}$;*Shh*$^{-/-}$ cell compartment, and that differences in Nkx2.2$^+$/Olig2$^+$ progenitors in that compartment would reflect Smo repression from Ptch1/2 in neighboring *Smo*$^{-/-}$ cells. We thus expected *Ptch1*$^{-/-}$;*Ptch2*$^{-/-}$;*Smo*$^{-/-}$ cells to have no effect. nEB mosaics of *Ptch1*$^{LacZ/LacZ}$;*Ptch2*$^{-/-}$;*Shh*$^{-/-}$ and *Ptch1*$^{-/-}$;*Ptch2*$^{-/-}$;*Smo*$^{-/-}$ co-cultured at 1:1 ratios met expectations by containing half as many Nkx2.2$^+$/Olig2$^+$ progenitors as *Ptch1*$^{-/-}$;*Ptch2*$^{-/-}$;*Shh*$^{-/-}$ nEBs. This indicates a lack of Ptch1/2-mediated Smo repression in the tissue (*Figure 2B*). In contrast, co-culture of *Ptch1*$^{LacZ/LacZ}$;*Ptch2*$^{-/-}$;*Shh*$^{-/-}$ cells with *Smo*$^{-/-}$ cells significantly decreased Nkx2.2$^+$ cells and increased Olig2$^+$ cells, rejecting the null hypothesis. We interpret this as a dorsal shift in the identity of the *Ptch1*$^{LacZ/LacZ}$;*Ptch2*$^{-/-}$;*Shh*$^{-/-}$ cell compartment, as Nkx2.2 is a marker of a more ventral neural progenitor domain than Olig2 (*Figure 2—figure supplement 1*). We attribute this relatively mild effect to low levels of Ptch1/2 activity in *Smo*$^{-/-}$ cells.

## Gene editing in a Disp1$^{-/-}$ background reveals non-cell autonomous regulation of Smo-mediated Ptch1:LacZ expression in mosaic nEBs

As an independent and more rapid assay for non-cell autonomous effects of Ptch1/2 on Smo we assessed Ptch1:LacZ induction (*Goodrich et al., 1997*) in nEBs using *Ptch1*$^{+/LacZ}$ and *Ptch1*$^{LacZ/LacZ}$ mESCs, and edited cell lines derived from them. Before employing this assay in mosaic nEBs, we investigated whether Ptch1:LacZ induction mirrored ventral neural progenitor differentiation (*Figure 1*) in our cell line panel. At 72 hr, Ptch1:LacZ levels were low in *Ptch1*$^{+/LacZ}$ and *Ptch1*$^{+/LacZ}$;*Shh*$^{-/-}$ nEBs but robust in *Ptch1*$^{LacZ/LacZ}$ and *Ptch1*$^{LacZ/LacZ}$;*Ptch2*$^{-/-}$ nEBs (*Figure 3A*). Ptch1:LacZ levels were reduced in *Ptch1*$^{LacZ/LacZ}$;*Shh*$^{-/-}$ nEBs but remained elevated in *Ptch1*$^{LacZ/LacZ}$;*Ptch2*$^{-/-}$;*Shh*$^{-/-}$ nEBs. *Ptch1*$^{LacZ/LacZ}$;*Smo*$^{-/-}$ *Ptch1*$^{LacZ/LacZ}$;*Ptch2*$^{-/-}$;*Smo*$^{-/-}$ nEBs had reduced Ptch:LacZ despite Ptch1/2 loss. *Smo*$^{-/-}$ mESCs were derived previously and constitutively express Rosa26:LacZ (*Zhang et al., 2001*). These measurements corroborate our findings in monotypic nEBs with neural progenitor markers and support a role for Smo-dependent, Shh ligand-mediated signaling in nEBs lacking Ptch1 (but not Ptch2), and the loss of Shh dependence in the complete absence of Ptch1/2 (*Figure 3A*). These

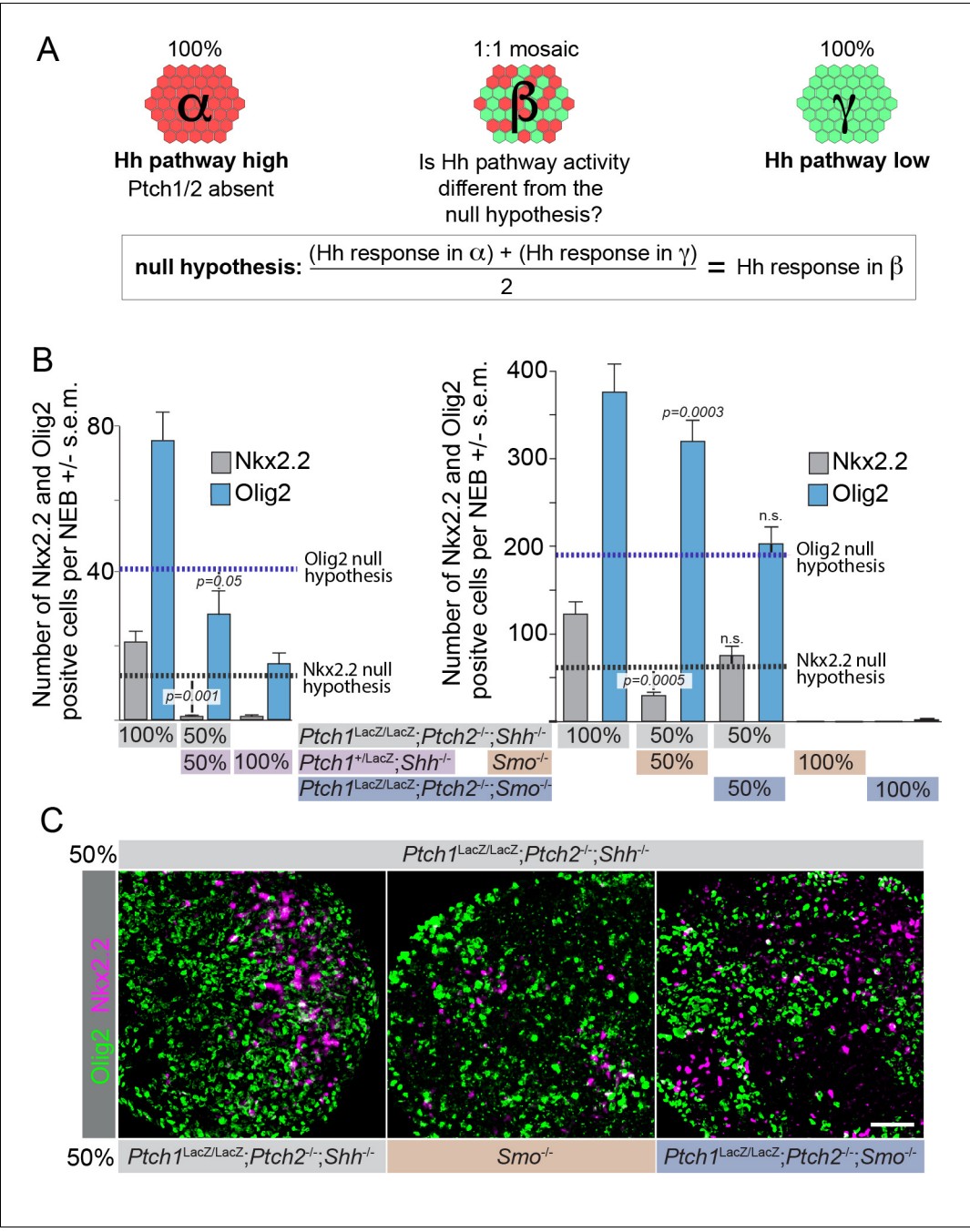

**Figure 2.** The Hh response in cells lacking Ptch1/2 can be inhibited non-cell autonomously by Ptch1/2 competent cells. (**A**) Diagram showing the experimental approach. Rejection of the null hypothesis provides evidence for non-cell autonomous activity of Ptch1/2. (**B**) Mosaic nEBs consisting 1:1 of $Ptch1^{+/LacZ};Shh^{-/-}$ and $Ptch1^{LacZ/LacZ};Ptch2^{-/-};Shh^{-/-}$ cells have fewer Nkx2.2+ and Olig2+ cells than predicted based on unmixed $Ptch1^{+/LacZ};Shh^{-/-}$ and $Ptch1^{LacZ/LacZ};Ptch2^{-/-};Shh^{-/-}$ nEBs (null hypothesis values, dotted lines). (**C**) Representative images of **B**, Scale bar is 50 µm. Two biological replicates were performed in **B**.
The following figure supplement is available for figure 2:

**Figure supplement 1.** Acquisition of a more dorsal identity in $Ptch1^{LacZ/LacZ};Ptch2^{-/-};Shh^{-/-}$ cells results in an increase in Olig2+ cells at the expense of Nkx2.2+ cells.

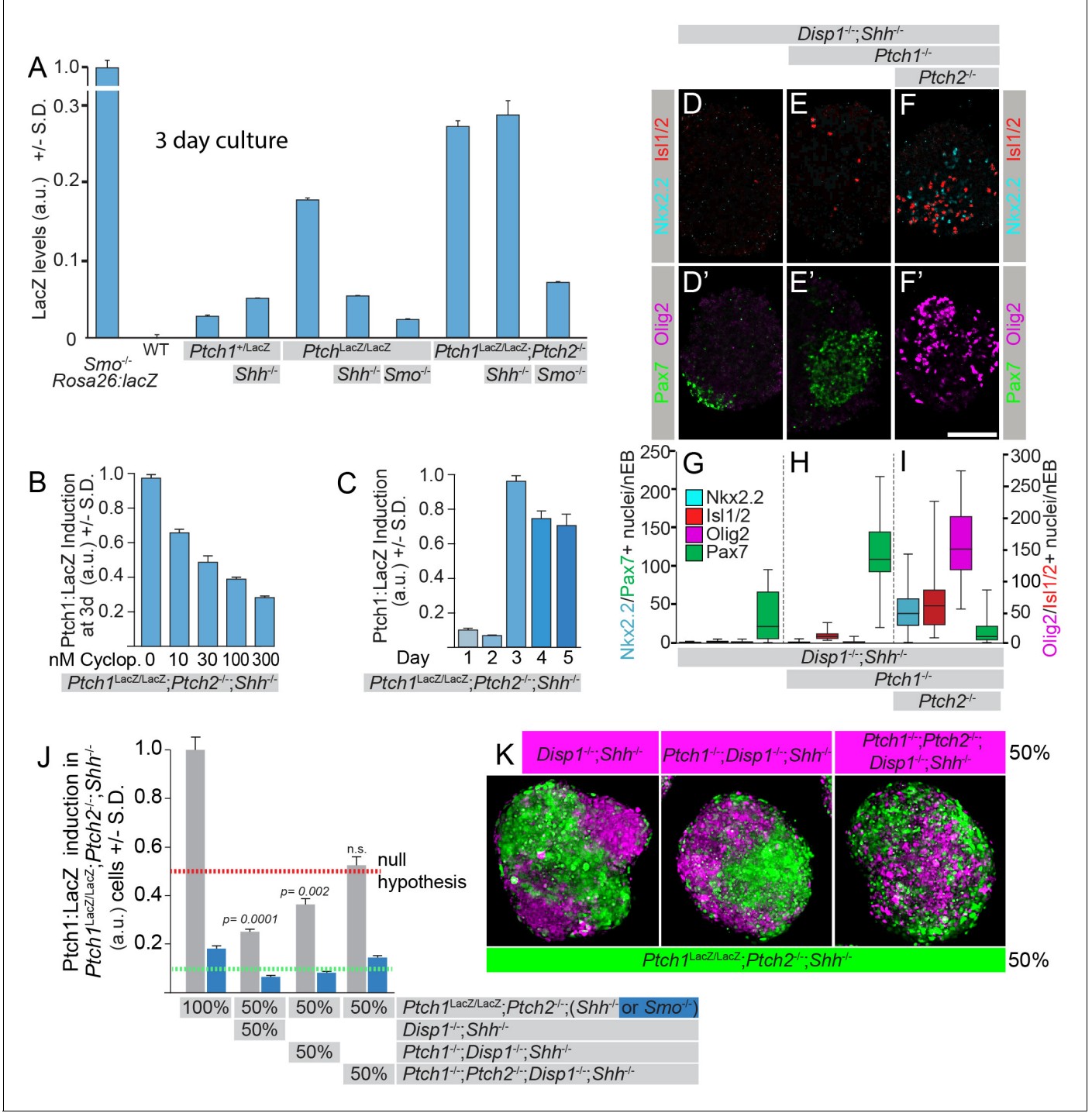

**Figure 3.** Ptch1/2 inhibit Ptch1:LacZ expression both cell autonomously and non-cell autonomously. (A) Ptch1:LacZ is a lineage-restricted measure of *Ptch1* expression, and thus Hh pathway activity. *Ptch1:LacZ* is activated by sequential loss of Ptch1/2, and this activation requires Smo activity. Genetic loss of *Shh* reduces the level of Hh pathway activation in *Ptch1*^LacZ/LacZ, but not in *Ptch1*^LacZ/LacZ;*Ptch2*^-/- cells (B) *Ptch1*^LacZ/LacZ;*Ptch2*^-/-;*Shh*^-/- nEBs were cultured in 0–300 nM cyclopamine and Ptch1:LacZ was measured at 72 hr. Ptch1:LacZ levels in 30 nM cyclopamine were approximately half those of untreated nEBs. (C) Ptch1:LacZ levels were measured up to 5 days after nEB formation. (D–I) Additional loss of *Disp1* does not alter the consequences of the loss of Ptch1/2 activity on neural progenitor identity. (J) Ptch1:LacZ expression in mosaic nEBs consisting of 50% *Ptch1*^LacZ/LacZ;*Ptch2*^-/-;*Shh*^-/- cells and 50% *Disp1*^-/-;*Shh*^-/-, *Disp1*^-/-;*Shh*^-/-;*Ptch1*^-/-, or *Disp1*^-/-;*Shh*^-/-;*Ptch1*^-/-;*Ptch2*^-/- cells (indicated). Ptch1:LacZ levels in the *Ptch1*^LacZ/LacZ;*Ptch2*^-/-;*Shh*^-/- cells were assessed after 72 hr in culture. LacZ levels were compared to half (red dotted line, signifying the null hypothesis) the LacZ activity measured in nEBs comprised of 100% *Ptch1*^LacZ/LacZ;*Ptch2*^-/-;*Shh*^-/- cells. Error bars are s.d., p-values are indicated (t-test, n = 3 measurements) and relate to comparison of each value with the null hypothesis. *Ptch1*^LacZ/LacZ;*Ptch2*^-/-;*Smo*^-/- cells have a low Ptch1:LacZ activity in this assay regardless of the

*Figure 3 continued on next page*

*Figure 3 continued*

genotype of the surrounding cells. (K) *Ptch1*$^{LacZ/LacZ}$;*Ptch2*$^{-/-}$;*Shh*$^{-/-}$ mESCs were loaded with green fluorescent cell tracker dye, and the *Disp1*$^{-/-}$;*Shh*$^{-/-}$, *Disp1*$^{-/-}$;*Shh*$^{-/-}$;*Ptch1*$^{-/-}$, and *Disp1*$^{-/-}$;*Shh*$^{-/-}$;*Ptch1*$^{-/-}$;*Ptch2*$^{-/-}$ mESCs with blue fluorescent dye (magenta). Mosaic nEBs as described in **A** were cultured for 48 hr, and imaged. 1:1 contribution reflecting the initial mosaic contributions is maintained throughout the experiment. *p* values are indicated (t-test). **D–I** are box-and-whisker plots.

data suggest that Ptch1:LacZ is a reliable output for Hh pathway activity across cell lines. We performed a time-course experiment to more precisely establish the point of maximal pathway activation. We found that Ptch1:LacZ was strongly induced in *Ptch1*$^{LacZ/LacZ}$;*Ptch2*$^{-/-}$;*Shh*$^{-/-}$ nEBs after 72 hr differentiation (*Figure 3C*), and this high level of expression persisted for several days. Ptch1:LacZ expression is suppressed after treatment with cyclopamine, indicating that Ptch1:LacZ upregulation requires Smo (*Figure 3B*).

To assess non-cell autonomous Ptch1/2 activity in a mosaic nEB assay using Ptch1:LacZ, we expanded our panel of mutant cell lines to include cells variable for Ptch1/2 status in *Disp1*$^{-/-}$ mESCs (*Etheridge et al., 2010*). We observed robust ventral neural progenitor identity only in *Disp1*$^{-/-}$; *Ptch1*$^{-/-}$;*Ptch2*$^{-/-}$;*Shh*$^{-/-}$ nEBs (*Ptch1*$^{LacZ}$ and *Ptch1*$^{-}$ distinguish null *Ptch1* alleles in these two families of cell lines). By contrast *Disp1*$^{-/-}$;*Shh*$^{-/-}$ and *Disp1*$^{-/-}$;*Ptch1*$^{-/-}$;*Shh*$^{-/-}$ had widespread Pax7$^{+}$ progenitors, indicating a low level of Hh pathway activation (*Figure 3D–I*).

*Disp1*$^{-/-}$ mESCs are devoid of LacZ and in co-cultures we can, therefore, strictly measure cell-non autonomous Smo regulation in *Ptch1*$^{LacZ/LacZ}$;*Ptch2*$^{-/-}$;*Shh*$^{-/-}$ cells. Additionally, using *Disp1*$^{-/-}$ cells in this assay ensures that Dhh and Ihh cannot compensate for *Shh* ablation, because all Hh ligands require Disp1 to mediate paracrine effects (*Etheridge et al., 2010*; *Ma et al., 2002*). Nevertheless, we genetically inactivated Shh in these cell lines as an additional safeguard against possible juxtacrine signaling by Shh (*Burke et al., 1999*; *Etheridge et al., 2010*; *Tsiairis and McMahon, 2008*).

We measured Ptch1:LacZ expression in mosaic nEBs at 72 hr when Ptch1:LacZ measurement reaches its maximum (*Figure 3C*). *Ptch1*$^{LacZ/LacZ}$;*Ptch2*$^{-/-}$;*Shh*$^{-/-}$ nEBs mosaic 1:1 with *Disp1*$^{-/-}$;*Shh*$^{-/-}$; *Ptch1*$^{-/-}$;*Ptch2*$^{-/-}$ cells yielded Ptch1:LacZ signal closely reflecting the relative mESC contribution of *Ptch1*$^{LacZ/LacZ}$;*Ptch2*$^{-/-}$;*Shh*$^{-/-}$ cells. However, when *Disp1*$^{-/-}$;*Shh*$^{-/-}$ or *Disp1*$^{-/-}$;*Shh*$^{-/-}$;*Ptch1*$^{-/-}$ cells were co-cultured with *Ptch1*$^{LacZ/LacZ}$;*Ptch2*$^{-/-}$;*Shh*$^{-/-}$ cells, Ptch1:LacZ levels in the *Ptch1*$^{LacZ/LacZ}$;*Ptch2*$^{-/-}$; *Shh*$^{-/-}$ cells significantly declined (*Figure 3J*). By contrast, only small differences were found in Ptch1: LacZ signal derived from *Ptch1*$^{LacZ/LacZ}$;*Ptch2*$^{-/-}$;*Smo*$^{-/-}$ mESCs co-cultured 1:1 with *Disp1*$^{-/-}$;*Shh*$^{-/-}$, *Disp1*$^{-/-}$;*Shh*$^{-/-}$;*Ptch1*$^{-/-}$ or *Disp1*$^{-/-}$;*Shh*$^{-/-}$;*Ptch1*$^{-/-}$;*Ptch2*$^{-/-}$ cells. This supports the notion that Smo is the target of the inhibitory cargo of Ptch1/2 activity. Labeling the two cell compartments using Cell Tracker dyes, and assessing the resulting mosaic nEBs showed equivalent relative contributions after 48 hr (*Figure 3K*), indicating that changes in LacZ levels in mosaic nEBs are not attributable to disparities in cell growth or adhesion during the culture period. These results suggest that endogenous Ptch1/2 suppress the Hh response non-cell autonomously in cells lacking Ptch1/2 activity.

## The loss of 7dhcr enhances the ability of cells to inhibit the Hh response pathway non-cell autonomously

We assessed 7-dehydroxycholesterol (7DHC), or one of its derivatives (*Bijlsma et al., 2006*; *Sever et al., 2016*) as a candidate for the Ptch1/2 cargo mediating non-cell autonomous Smo inhibition. *7dhcr* mutations are associated with Shh signaling defects caused by the accumulation of a late sterol precursor (or its derivative) that inhibits Smo, according to the prevailing hypothesis (*Bijlsma et al., 2006*; *Cohen, 2010*; *Gruchy et al., 2014*; *Incardona et al., 2000a*). We thus tested whether loss of *7-Dehydroxycholesterol Reductase* (*7dhcr*) enhances Ptch1/2-mediated inhibition of Smo in adjacent *Ptch1*$^{LacZ/LacZ}$;*Ptch2*$^{-/-}$;*Shh*$^{-/-}$ cells.

Using CRISPR/Cas9 gene editing (*Doudna and Charpentier, 2014*), we inactivated *7dhcr* in previously edited *Shh*$^{-/-}$ mESCs devoid of LacZ, adding *Shh*$^{-/-}$;*7dhcr*$^{-/-}$ cells to our panel of mutant cell lines. We generated 1:1 mosaic nEBs consisting of *Ptch1*$^{LacZ/LacZ}$;*Ptch2*$^{-/-}$;*Shh*$^{-/-}$ cells and either *Shh*$^{-/-}$ or *Shh*$^{-/-}$;*7dhcr*$^{-/-}$ cells (*Figure 4A*) and measured Hh pathway activation exclusively in the *Ptch1*$^{LacZ/}$$^{LacZ}$;*Ptch2*$^{-/-}$;*Shh*$^{-/-}$ cell population via Ptch1:LacZ. Differences in Ptch:LacZ levels were thus attributable to differing degrees of cell non-autonomous regulation by the other cell compartment. Significantly less Ptch1:LacZ signal was obtained from nEBs containing *Shh*$^{-/-}$;*7dhcr*$^{-/-}$ cells as compared

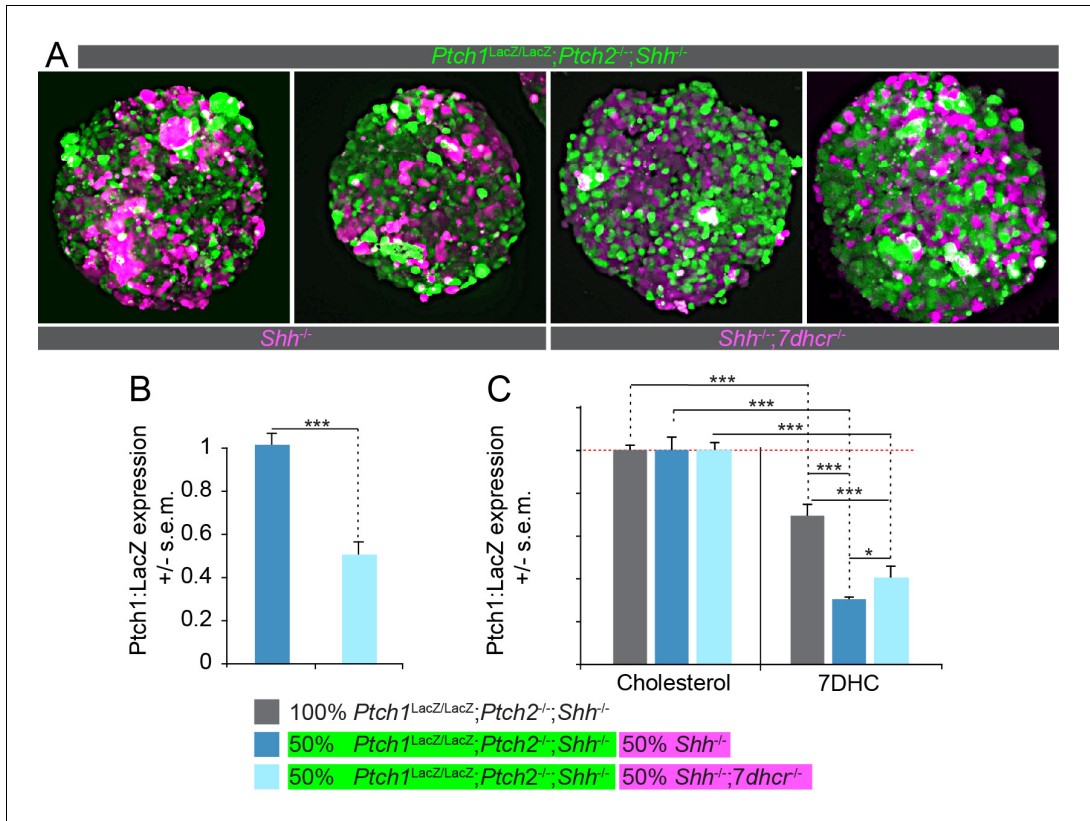

**Figure 4.** Loss of *7dhcr* enhances non-cell autonomous suppression of the Hh response. (**A**) nEBs consisting of 50% *Ptch1*$^{LacZ/LacZ}$;*Ptch2*$^{-/-}$;*Shh*$^{-/-}$ cells (green) and 50% *Shh*$^{-/-}$ cells or 50% *Shh*$^{-/-}$; *7dhcr*$^{-/-}$ cells (magenta) were labeled with cell tracking vital dyes. (**B**) LacZ quantification in nEBs described in (**A**). *7dhcr* ablation enhances non-cell autonomous inhibition of the Hh response in *Ptch1*$^{LacZ/LacZ}$;*Ptch2*$^{-/-}$;*Shh*$^{-/-}$ cells. (**C**) The Hh response in *Ptch1*$^{LacZ/LacZ}$;*Ptch2*$^{-/-}$;*Shh*$^{-/-}$ cells is inhibited by 7-Dehydrocholesterol (7DHC) as compared to cholesterol. The ability of 7DHC to inhibit the Hh response in *Ptch1*$^{LacZ/LacZ}$;*Ptch2*$^{-/-}$;*Shh*$^{-/-}$ cells is exacerbated by the inclusion of *Shh*$^{-/-}$ cells, and to a lesser extent by *Shh*$^{-/-}$;*7dhcr*$^{-/-}$ cells. n > 8, *p<0.05, ***p<0.001, n.s., not significant (t-test).

with nEBs containing parental *Shh*$^{-/-}$ cells (**Figure 4B**). Cell tracker dyes used to label each cell compartment indicated that all cell lines contributed to mosaic nEBs equally (**Figure 4A**). Because nEBs are cultured in a cholesterol-free environment, they likely upregulate cholesterol synthesis (**Brown and Goldstein, 1986**). Accumulation of 7DHC, or a 7DHC derivative (**Bijlsma et al., 2006**; **Sever et al., 2016**) in *Shh*$^{-/-}$;*7dhcr*$^{-/-}$ cells may thus provide a larger pool of the Ptch1/2 cargo, increasing secretion of the Smo-inhibitory sterol, possibly 7DHC.

To further test the role of 7DHC in Smo inhibition, we cultured *Ptch1*$^{LacZ/LacZ}$;*Ptch2*$^{-/-}$;*Shh*$^{-/-}$ nEBs in its presence. Corroborating earlier findings (**Bijlsma et al., 2006**), we observed a decreased Hh response in nEBs treated with 7DHC compared to control treatment with cholesterol, regardless of whether the nEB was mosaic or comprised exclusively of *Ptch1*$^{LacZ/LacZ}$;*Ptch2*$^{-/-}$;*Shh*$^{-/-}$ cells (**Figure 4C**). We additionally saw differences between the mosaic nEBs. A larger inhibition of the Hh response by exogenous 7DHC was observed in *Ptch1*$^{LacZ/LacZ}$;*Ptch2*$^{-/-}$;*Shh*$^{-/-}$ cells co-cultured 1:1 in nEBs with *Shh*$^{-/-}$ as opposed to *Shh*$^{-/-}$;*7dhcr*$^{-/-}$ cells (**Figure 4C**). We speculate that cells containing Ptch1/2 may be able to process or transport exogenous 7DHC into a more potent non-cell autonomous inhibitor of the Hh response in neighboring *Ptch1*$^{LacZ/LacZ}$;*Ptch2*$^{-/-}$;*Shh*$^{-/-}$ cells, and that this effect is further enhanced in cells genetically intact for *7dhcr* and presumably producing endogenous 7DHC, in addition to the exogenous source. These findings are consistent with Ptch1/2 antiporter activity mediating secretion of 7DHC, or an oxysterol derivative like 3β,5α-dihydroxycholest-7-en-6-one (**Sever et al., 2016**), as a mechanism to inhibit Smo in neighboring cells.

## Motor neuron differentiation in wild-type HB9:GFP cells is attenuated by Ptch1/2 in nearby cells via a SAG-competitive mechanism

Motor neurons arise from a population of neural progenitors in the ventral neural tube, and Hh pathway activation is required for motor neuron differentiation in vivo as well as in nEBs. We used the induction of GFP in HB9:GFP (*Wichterle et al., 2002*) mESCs as an independent lineage-restricted measure of the Hh response in genetically mosaic nEBs. HB9:GFP⁺ cells had motor neuron morphology and co-labeled with Isl1/2 immunostain, and thus serve as a measure for motor neuron induction (*Figure 5B*, *Figure 1—figure supplement 1*). Mosaic nEBs consisting of *Ptch1*$^{+/LacZ}$;*Shh*$^{-/-}$, *Ptch1*$^{LacZ/-}$;*Shh*$^{-/-}$ or *Ptch1*$^{LacZ/LacZ}$;*Ptch2*$^{-/-}$;*Shh*$^{-/-}$ mESCs mixed 10:1 with HB9:GFP mESCs were generated. We observed a small but significant increase in HB9:GFP⁺ motor neurons when these cells were co-cultured with *Ptch1*$^{LacZ/LacZ}$;*Ptch2*$^{-/-}$;*Shh*$^{-/-}$ cells as compared to *Ptch1*$^{+/LacZ}$;*Shh*$^{-/-}$ or *Ptch1*$^{LacZ/-}$;*Shh*$^{-/-}$ cells (*Figure 5A,C*).

While we hypothesize that Ptch1/2-expressing cells produce a Smo inhibitor, it remains a formal possibility that cells lacking Ptch1/2 have an activating effect on nearby cells. This activating activity could be indirect in that Shh potentially produced by HB9:GFP cells would no longer be sequestered by Ptch1/2 in adjacent cells (*Figure 1T*), making Shh available to the HB9:GFP cells themselves. To discriminate between these possibilities, we treated mosaic nEBs with SAG, a small molecule Smo agonist thought to antagonize the Smo-inhibitory Ptch1/2 substrate (*Chen et al., 2002a*; *Sharpe et al., 2015*). We expected decreased availability of the inhibitor, due to Ptch1/2 absence in neighboring cells, to enhance SAG effects. If cells lacking Ptch1/2 release an activator of the Hh response, its effects in combination with SAG are expected to be additive.

As predicted, 10 nM and 100 nM SAG induces motor neuron differentiation in HB9:GFP cells under all mosaic conditions (*Figure 5A*). However, a strong synergistic effect was observed between Ptch1/2 loss in surrounding cells and SAG-induced motor neuron differentiation in HB9:GFP cells (*Figure 5A,B*). We found that compared to *Ptch1*$^{LacZ/LacZ}$;*Ptch2*$^{-/-}$;*Shh*$^{-/-}$ cells, *Ptch1*$^{+/LacZ}$;*Shh*$^{-/-}$ cells suppressed motor neuron induction at both SAG concentrations, while *Ptch1*$^{LacZ/-}$;*Shh*$^{-/-}$ cells also suppressed motor neuron induction to an intermediate degree (presumably via Ptch2 activity). This observation is consistent with Ptch1/2 cargo acting as a SAG antagonist and is not easily reconciled with impaired Hh ligand sequestration due to the lack of Ptch1/2 in the environment, as this scenario should have little effect on SAG-mediated Smo activation.

## Three-part mosaic nEBs reveal a non-cell autonomous role for Ptch1/2 in regulating the response to Shh ligand

Shh activates Smo activity indirectly, unlike SAG, after first binding Ptch1/2, according to the canonical Hh signaling model (*Chen et al., 2002b*). To determine whether our previous findings apply to signaling by mature Shh ligand, we investigated whether Ptch1/2 within mosaic nEBs inhibit Smo activation in HB9:GFP cells in response to Shh expressed by a third cell population. To accomplish this, we generated three-part mosaic nEBs including 1% wild type cells harboring the EF1α:Shh transgene. These cells functioned as sparse, localized sources of Shh in mosaic nEBs. Effects of Shh produced by these cells were measured in HB9:GFP cells (5% of cells) in mosaic nEBs in which the *Ptch1/2* genotype in the third and predominant compartment (94% of cells) was varied.

Three-part mosaic nEBs consisting of 94% *Ptch1*$^{LacZ/LacZ}$;*Shh*$^{-/-}$ or *Ptch1*$^{LacZ/LacZ}$;*Ptch2*$^{-/-}$;*Shh*$^{-/-}$ cells facilitated robust Shh-mediated HB9:GFP⁺ motor neuron induction. In contrast, we observed negligible GFP expression in mosaic nEBs principally comprised of *Ptch1*$^{+/LacZ}$;*Shh*$^{-/-}$ cells (*Figure 6A,D*). HB9:GFP⁺ motor neurons were not observed when Shh overexpressing cells were omitted. Thus, the response to Shh is strongly enhanced by Ptch1/2 absence in nearby cells.

Our experiments with SAG (*Figure 5*) make it unlikely that lack of Hh ligand sequestration causes non-cell autonomous enhancement of the Hh response in nEBs lacking Ptch1/2. Nevertheless, we directly tested Shh abundance in mosaic nEBs. Live staining with the anti-Shh monoclonal antibody 5E1 is expected to exclusively bind Shh present in the extracellular space. Moreover, the 5E1 epitope on Shh overlaps with the binding site of Ptch1, preventing visualization of Ptch1/2 sequestered Shh (*Fuse et al., 1999*; *Pepinsky et al., 2000*). We found no difference in extracellular Shh staining in various mosaic nEBs (*Figure 6—figure supplement 1*), further supporting the idea that non-cell autonomous inhibition by Ptch1/2 of the Hh response does not involve Shh sequestration.

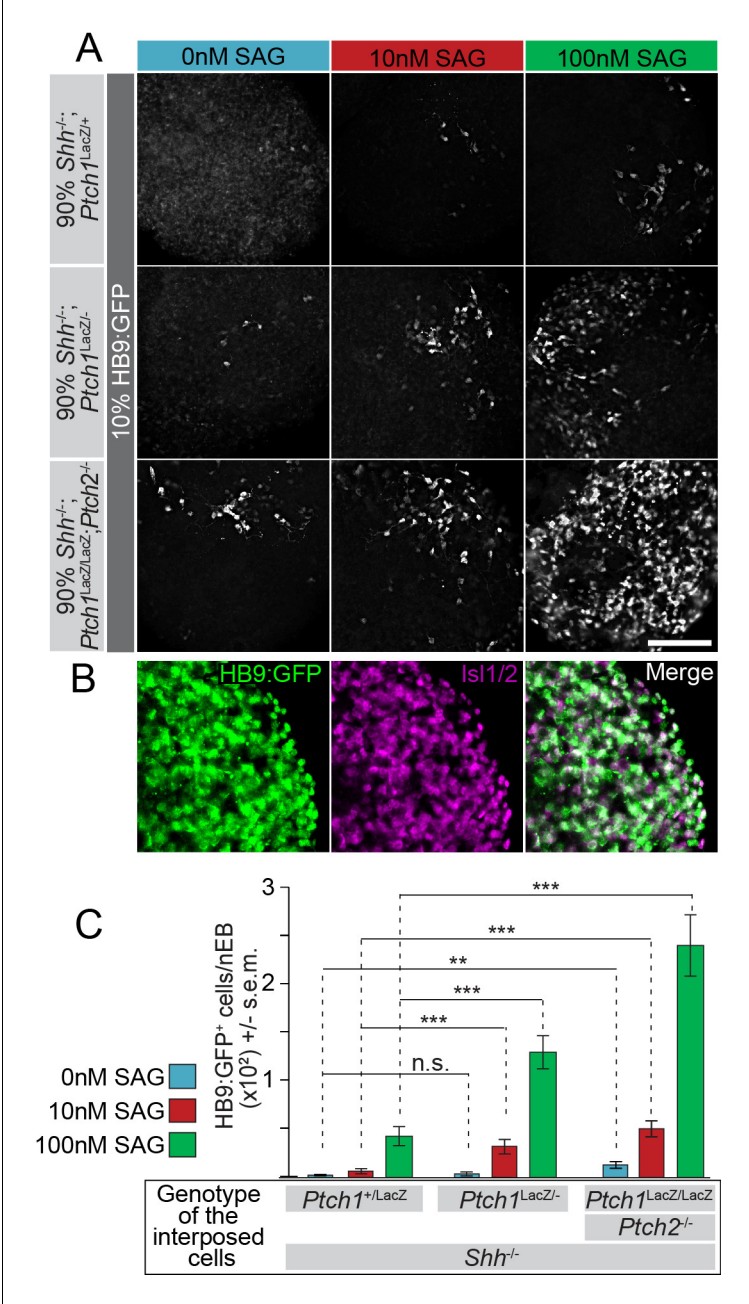

**Figure 5.** The Hh response to the Smo agonist SAG in HB9:GFP cells is enhanced by the absence of Ptch1/2 in neighboring cells. (**A**) Images of mosaic nEBs consisting of 10% HB9:GFP cells and 90% $Ptch1^{+/LacZ};Shh^{-/-}$, $Ptch1^{LacZ/-};Shh^{-/-}$ or $Ptch1^{LacZ/LacZ};Ptch2^{-/-};Shh^{-/-}$ cells (indicated). Mosaic nEBs were cultured in 0 nM (blue), 10 nM (red) or 100 nM (green) SAG. GFP expression in HB9:GFP cells indicates motor neuron differentiation, a measure of Hh pathway upregulation. (**B**) Isl1/2 (magenta) and HB9:GFP (green) is largely confined to the same cells, indicating that HB9:GFP serves as a motor neuron marker. (**C**) HB9:GFP+ cells were quantified. n > 20, **p<0.01, ***p<0.001, n.s., not significant (t-test). Scale bar is 100 μm for **A**, 200 μm for **B**.

To address if Ihh or Dhh explained enhanced motor neuron induction in mosaic nEBs, we varied Ptch1/2 status in $Disp1^{-/-};Shh^{-/-}$ cells, in which Ihh and Dhh are not expected to signal. Shh robustly induced HB9:GFP+ motor neurons in nEBs consisting of 94% $Disp1^{-/-};Ptch1^{-/-};Ptch2^{-/-};Shh^{-/-}$ cells, while motor neuron induction was negligible in nEBs primarily consisting of 94% $Disp1^{-/-};Ptch1^{-/-};$

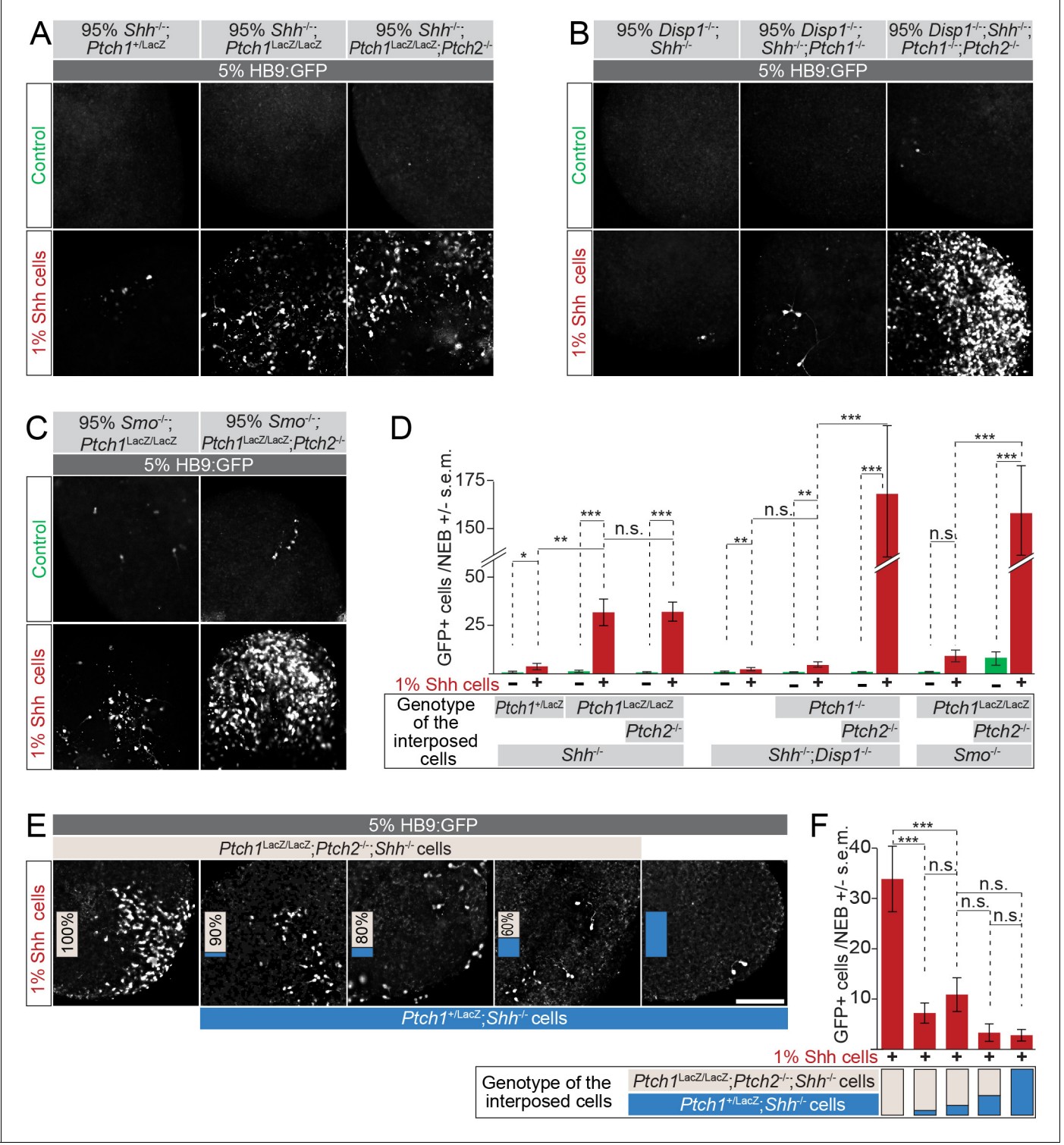

**Figure 6.** Loss of Ptch1 and Ptch2 in neighboring cells enhances the response to Shh in HB9:GFP cells. (A–C) Images of two-part and three-part mosaic nEBs showing GFP expression in HB9:GFP cells. All nEBs included 5% HB9:GFP cells, and 1% Shh-expressing cells where indicated. (A) The remaining 94%/95% of cells were $Ptch1^{+/LacZ};Shh^{-/-}$, $Ptch1^{LacZ/LacZ};Shh^{-/-}$ or $Ptch1^{LacZ/LacZ};Ptch2^{-/-};Shh^{-/-}$ (indicated). (B) The remaining 94%/95% of cells were $Disp1^{-/-};Shh^{-/-}$, $Disp1^{-/-};Shh^{-/-};Ptch1^{-/-}$, or $Disp1^{-/-};Shh^{-/-};Ptch1^{-/-}Ptch2^{-/-}$ (indicated). (C) Remaining 94%/95% of cells were $Ptch1^{-/-};Smo^{-/-}$ or $Ptch1^{-/-};Ptch2^{-/-};Smo^{-/-}$ (indicated). Under all conditions, Ptch1/2 absence greatly enhanced Shh-dependent motor neuron differentiation in HB9:GFP cells. (D) HB9:GFP$^{+}$ cells in (A), (B) and (C) were quantified per mosaic nEB. (E) Images of three-part and four-part mosaic nEBs showing HB9:GFP$^{+}$ cells. All nEBs included 5% HB9:GFP cells and 1% Shh-expressing cells. Remaining cells were $Ptch1^{LacZ/LacZ};Ptch2^{-/-};Shh^{-/-}$ (gray) and $Ptch1^{+/LacZ};Shh^{-/-}$ (blue) in

*Figure 6 continued on next page*

Figure 6 continued

indicated ratios. *Ptch1*<sup>+/LacZ</sup>;*Shh*<sup>-/-</sup> cells suppress Shh-mediated motor induction. (**F**) HB9:GFP$^+$ cells in (**E**) were quantified. *p<0.05, **p<0.01, ***p<0.001, n.s., not significant (t-test). Scale bar is 100 μm.

The following figure supplement is available for figure 6:

**Figure supplement 1.** Shh distribution away from the sites of synthesis is not affected by the absence of Ptch1/2 in the tissue.

*Shh*$^{-/-}$ and *Disp1*$^{-/-}$;*Shh*$^{-/-}$ cells (*Figure 6B,D*). The degree of non-cell autonomous repression of HB9:GFP induction in response to Shh reflects the extent of dorsal identity, and presumably Ptch1/2 activity, in this family of cell lines (*Figure 3D–I*).

The activated Hh pathway in Ptch1/2-deficient cells could cause motor neuron induction via unknown downstream factors. To address this, we generated three-part mosaic nEBs using *Ptch1*$^{-/-}$; *Smo*$^{-/-}$ or *Ptch1*$^{-/-}$;*Ptch2*$^{-/-}$;*Smo*$^{-/-}$ mESCs. HB9:GFP$^+$ motor neurons were induced efficiently only in predominantly *Ptch1*$^{-/-}$;*Ptch2*$^{-/-}$;*Smo*$^{-/-}$ nEBs (*Figure 6C,D*). This observation again supports overlapping non-cell autonomous roles for Ptch1/2.

### The inhibitory effects of Ptch1/2 containing cells are dominant over those of cells lacking Ptch1/2 function in four-part mosaic nEBs

To further exclude the possibility that cells lacking Ptch1/2 produce a Hh pathway inducer, we tested if the non-cell autonomous properties of cells lacking Ptch1/2 are dominant over cells with Ptch1/2 function (suggesting secretion of an inducer) or if the activities associated with Ptch1/2 containing cells are dominant over those lacking Ptch1/2 function (supporting secretion of an inhibitor). We made nEBs that were largely composed of various ratios of *Ptch1*<sup>LacZ/LacZ</sup>;*Ptch2*$^{-/-}$;*Shh*$^{-/-}$ and *Ptch1*<sup>+/LacZ</sup>;*Shh*$^{-/-}$ cells, and assessed their effect on the induction of HB9:GFP$^+$ cells by Shh-expressing cells. In nEBs largely comprised of *Ptch1*<sup>LacZ/LacZ</sup>;*Ptch2*$^{-/-}$;*Shh*$^{-/-}$ cells, even a minor fraction of *Ptch1*<sup>+/LacZ</sup>; *Shh*$^{-/-}$ cells suppressed motor neuron induction (*Figure 6E,F*). This indicates that the properties of Ptch1/2 containing cells are dominant over those of cells lacking Ptch1/2 function, further supporting the notion that Ptch1/2 function mediates the secretion of a Smo inhibitor that de-sensitizes cells to the effects of Shh.

### Discussion

Mosaic nEBs comprised of mESCs with novel, complex genotypes allow us to study interactions between cell populations with resolution not easily achieved in vivo. Within this system, lineage-restricted reporter cells unambiguously indicate non-cell autonomous Smo inhibition by nearby cells expressing Ptch1/2 at endogenous levels. A logical interpretation of these results is that Ptch1/2 mediate the secretion of a Smo inhibitor that affects the Hh response both cell autonomously, and in nearby cells.

Observations similar to ours but using overexpressed Ptch1 have been reported in fibroblasts and our experiments support this finding (*Bijlsma et al., 2006*). Detecting these activities of Ptch1/2 expressed at endogenous levels in nEBs resolves the argument that overexpressed Ptch1/2 constructs could have non-physiological effects. Because Shh signaling in nEBs patterns multiple cell fates and mimics neural tube induction, these results are likely relevant to in vivo signaling.

Although all independently derived Ptch1/2 cells were equally unable to inhibit the Hh response non-cell autonomously, we found that independent *Ptch1*$^{-/-}$ lines varied in regard to this activity. Notably, cells with a dorsal identity (*Disp1*$^{-/-}$;*Ptch1*$^{-/-}$;*Shh*$^{-/-}$) are better non-cell autonomous repressors of the Hh response than cells with a more ventral identity (*Ptch1*<sup>LacZ/LacZ</sup>;*Shh*$^{-/-}$). We speculate that this is due to varying levels of Ptch2 activity. It also appears that the loss of Smo decreases the ability of Ptch1/2 to inhibit the Hh response non-cell autonomously. This observation is easily explained by a decrease in Ptch1/2 levels as Ptch1/2 expression is under the control of Smo. Thus, even in the complete absence of Hh pathway activation, Ptch1/2 can still inhibit Smo. This is consistent with the observation that in tissues without detectable levels of Ptch1/2, Smo remains inactive.

Our findings are consistent with Ptch1/2 functioning as proton-driven efflux pumps in the RND family (*Nikaido and Takatsuka, 2009*). RND antiporters utilize a pH gradient to drive transport and

Ptch1/2 thus likely function in acidified compartments. We previously demonstrated that Ptch1 localizes to late endosomes while mediating Shh uptake, and that Shh signaling requires endosomal acidification (*Incardona et al., 2002*). Conserved acidic residues required for proton flux in prokaryotic RNDs are also required for Ptch1 to repress Smo (*Alfaro et al., 2014*; *Strutt et al., 2001*). One possibility is that Ptch1/2 enrich the endosomal lumen or intraluminal vesicular in multivesicular endosomes (MVEs) with a Smo inhibitor. Exosomal release would allow this inhibitor to enter the extracellular environment and regulate Smo both cell autonomously as well as non-cell autonomously.

Our observation that cells enriched in sterol precursors are better cell non-cell autonomous inhibitors of the Hh response complements earlier observations implicating sterols as the Smo inhibitors transported by Ptch1/2. The observation that 7DHCR loss coincides with reduced Hh signaling is refined by our results showing that Ptch1/2 become more effective non-cell autonomous inhibitors of Smo in nearby cells when expressed in cells enriched for 7DHC or its derivatives.

Why the Smo-inhibitory Ptch1/2 cargo, despite its likely abundance in cells, fails to inhibit Smo without being acted on (cell autonomously or non-cell autonomously) by Ptch1/2 remains unresolved. However, our results show that 7DHC becomes a more potent inhibitor when acted upon by Ptch1/2 activity, and when combined with the evidence that Ptch1/2 function as proton-driven antiporters, it becomes plausible that the Ptch1/2 cargo becomes inhibitory after translocation or secretion. This view is consistent with the known role of the prokaryotic RND HpnN (*Doughty et al., 2011*) that transports bacterial sterols between inner and outer membranes. It would also be consistent with the function of NPC1, a close relative of Ptch that translocates cholesterol between intracellular membranes (*Blanchette-Mackie, 2000*).

Ptch1/2 mutations drive the formation of several tumors, and an important ramification of our findings is that Ptch1/2 disruption enhances not only cell autonomous Hh responses, but also Smo activation in adjacent cells with intact Ptch1/2 activity (*Barakat et al., 2010*). The finding that genetically normal stromal cells respond to Shh expressing tumors by infiltrating and supporting them heightens the importance of our observations because Ptch1/2 loss in the tumor may affect Shh sensitivity in supporting stromal cells non-cell autonomously (*Yauch et al., 2008*). Our results also predict that even in the absence of Ptch1/2, cells remain sensitive to Hh ligands signaling in nearby cells. Anticancer strategies based on ligand sequestration or inactivation therefore remain viable treatment options.

Hh signaling plays many critical roles during development as a morphogen. Responding cells interpret graded Hh ligand distributions, resulting in stereotyped patterning, and Ptch1/2 have complex roles in this process. As Hh receptors, Ptch1/2 bind extracellular Shh and initiate the response. In Drosophila, Ptch activation results in the accumulation of phosphatidylinositol 4-phosphate that in turn activates Smo via its intracellular C-terminal domain (*Jiang et al., 2016*; *Yavari et al., 2010*), a mechanism conserved in vertebrates (*Jiang et al., 2016*). Invariably, Hh signaling induces *Ptch1/2* expression (*Holtz et al., 2013*) and Ptch1/2 induction then leads to negative feedback, possibly by secreting more Smo inhibitor, increasing Shh sequestration, or both. Our finding that Ptch1/2 inhibit the Hh response non-cell autonomously, even in nEBs devoid of Shh ligand, supports the notion that the non-cell autonomous inhibition mediated by Ptch1/2 is mediated by the antiporter activity of Ptch1/2, rather than by ligand sequestration. The ability of relatively few Ptch1/2 expressing cells to inhibit the Hh response pathway further supports this idea.

Together these findings indicate that Ptch1/2 act broadly and communally inhibit Smo in tissues undergoing patterning. According to this model, local sensitivity to Shh is highly buffered and equalized between cells, aiding the formation of a smooth response gradient in the Shh morphogenetic field.

## Materials and methods

### Cell lines

*Ptch1*[+/LacZ] and *Ptch1*[LacZ/LacZ] mESCs were gifts from Dr. Matthew Scott (Stanford University and HHMI). Identity of these lines was confirmed by the presence of the *LacZ* recombination in the *Ptch1* locus, the presence of 40 chromosomes per cell, and mouse-specific DNA sequences of the edited genes. *Smo*[-/-] mESCs were a gift from Dr. Andrew McMahon (University of Southern California), and

their identity was confirmed by the presence of the *ROSA26:LacZ* locus and the absence of *Smo*. HB9:GFP mESCs were a gift from Dr. Thomas Jessell (Columbia University). Their identity was confirmed by the presence of the *Hb9:gfp* transgene. *Disp1*[-/-] mESCs and wild type (AB1) mESCs, and mESCs overexpressing Shh were previously described (*Etheridge et al., 2010*). Identity of these lines was confirmed by the presence of 40 chromosomes per cell, and mouse-specific DNA sequences of the edited genes. mESC lines were maintained using standard conditions in dishes coated with gelatin, without feeder cells. Cells were routinely tested for Mycoplasm by Hoechst stain, and grown in the presence of tetracycline and gentamycin at regular intervals. Cultures with visible Mycoplasma infection were discarded. None of the cell lines used in this study is listed in the Database of Cross-Contaminated or Misidentified Cell Lines.

## Materials
Cyclopamine was a gift from Dr. William Gaffield (USDA) (*Gaffield and Keeler, 1996*). SAG was from EMD Biochemicals (Darmstadt, Germany). Retinoic acid was from Sigma/Aldrich (St. Louis, MO).

## Immunostaining
Mouse anti-Pax7 (RRID:AB_528428), anti-Pax6 (RRID:AB_528427) and anti-Nkx2.2 (RRID: AB_2314952 AB_531794), were obtained from the Developmental Studies Hybridoma Bank. Goat anti-Olig2 (RRID: AB_2157554) was purchased from R&D Systems (Minneapolis, MN). Guinea pig anti-Isl1/2 was a gift from Dr. Thomas Jessell (Columbia University). In all experiments, donkey and goat Alexa-488 anti-mouse, goat Alexa-568 anti-guinea pig and donkey Alexa-568 anti-goat were used as secondary antibodies. nEBs were mounted in Fluormount-G and positive nuclei were quantified. Fixation was performed for 10 min on ice using 4% paraformaldehyde in 1X PBS. Native HB9:GFP fluorescence was imaged directly, after fixation and mounting, without antibody detection.

## Imaging and quantification of nuclear progenitor markers
Mounted nEBs were imaged with a Zeiss Observer fluorescence microscope with a 20x objective. Within each experiment, stacks were de-convolved and resulting image files were scrambled for unbiased, blind counting.

## Fluorescent tracking of cells
Cells were singularized and washed twice with PBS. Cells were stained with 20 µM CellTracker Blue CMAC or CellTracker Green CMFDA (Thermo Fisher Scientific, Waltham MA) in DFNB for 45 min. The cells were mixed as described above, incubated at 37°C with agitation (~0.8 Hz) for 48 hr, fixed in 4% PFA for 10 min, and mounted for microscopy. Signal was insufficient at 72 hr.

## Neuralized embryoid body differentiation
mESCs were differentiated into nEBs using established procedures (*Wichterle et al., 2002*). nEBs were aggregated for 48 hr in DFNB medium in Petri dishes rotated at 0.8 Hz. 2 µM Retinoic Acid (RA) was added at 48 hr. nEBs were fixed 48 hr after the addition of RA for antibody staining of neural progenitors. nEBs were fixed 96 hr after the addition of RA for imaging and quantifying HB9:GFP fluorescence.

## Reporter gene assay for Ptch1:LacZ activity
nEBs were collected, washed once in PBS and lysed in 100 mM potassium phosphate, pH 7.8, 0.2% Triton X-100. Lysates were analyzed using the Galacto-Light chemiluminescent kit (Applied Biosciences, Foster City, CA) for Ptch1:LacZ expression level. Lysates were normalized for total protein using the Bradford reagent (BioRad, Hercules, CA). At least three technical replicates are reported for each measurement.

## Genome editing
TALEN constructs, transfection, mESC clone selection and genotyping, and domain architectures for TALEN constructs targeting *Shh* and *Ptch2* were previously described (*Alfaro et al., 2014*). *Ptch1* and *Smo* were targeted similarly. We repeated the protocol sequentially in mESC lines to generate

complex mutant genotypes. Repeat variable domain architectures for TALEN constructs targeting *Ptch1* and *Smo* were: *Ptch1* (5′) NN NN HD HD NG HD NN NN HD NG NN NN NG NI NI and (3′) HD HD HD NN HD HD NN HD HD NN NN HD HD NG NN HD HD NG NN *Smo* (5′) NN HD NG NN HD NG NN NN NG NI HD NG NN HD NG and (3′) HD HD HD NN HD NG HD NI NI NN NN HD HD NN HD HD HD.

sgRNAs were designed using the online CRISPR Design tool (http://tools.genome-engineering. org). Duplexed oligos (5′) CACCGCTGTCGCTGTGATGAACACG (3′), (5′) CACCGAACACG TGGCAAAAAGCAGC (3′) and (5′) CACCGAATGAACTACAATTCGGAAT(3′), (5′) CACCGGCTCAC-CAGTGACCCTTATC (3′) bracketing exon 1 and exon 2 of mouse *7dhcr* were cloned into pX459 (*Ran et al., 2013*). *Shh*⁻/⁻ mESCs were transfected and transiently selected with puromycin and individual clones isolated.

## Genotyping

PCR screening for *Ptch2* and *Shh* was previously described (*Alfaro et al., 2014*). PCR screening was performed on cell lysates using primers flanking the *Ptch1* and *Smo* TALEN binding sites: *Ptch1*: (5′) GCAAAGACCTCGGGACTCA (3′) and (5′) GGAGGGAGGGTTTGAATTTTT (3′). *Smo*: (5′) GCACCGG TCGCCTAAGTAGC (3′) and (5′) GCACACGTTGTAGCGCAAA (3′). Deletions of *7dhcr* sequences were confirmed by PCR using primers (5′) AGATCCTGCACAAAGCGCAC (3′) and (5′) ACAGGC TGAGTAAGCCTTCAAGC (3′) amplifying a 300 bp fragment after the anticipated edit, and absence of a PCR product using (5′) TCAGTCTCACGCAGCAAGCAG (3′) and (5′) AGATCCTGCACAAAGCG-CAC (3′). Individual clones were confirmed by sequencing.

## Mutations

*Ptch1*^LacZ/LacZ^;*Ptch2*⁻/⁻ and *Ptch1*^LacZ/LacZ^;*Shh*⁻/⁻ mESCs were previously described (*Alfaro et al., 2014*). *Ptch1*^+/LacZ^;*Shh*⁻/⁻ were heterozygous for 1 bp and 10 bp deletions in *Shh* exon 1. *Ptch1*^LacZ/LacZ^;*Ptch2*⁻/⁻;*Shh*⁻/⁻ mESCs were heterozygous for a 1 bp insertion and a 4 bp deletion in *Shh* exon 1. *Ptch1*^LacZ/LacZ^;*Smo*⁻/⁻ were heterozygous for 90 bp and 110 bp deletions in *Smo* exon 1. *Ptch1*^LacZ/LacZ^;*Ptch2*⁻/⁻;*Smo*⁻/⁻ mESCs were homozygous for an 83 bp deletion in *Smo* exon 1. *Disp1*⁻/⁻;*Shh*⁻/⁻ were heterozygous for 16 bp and 35 bp deletions in *Shh* exon 1. *Disp1*⁻/⁻;*Shh*⁻/⁻;*Ptch1*⁻/⁻ mESCs were additionally homozygous for a 1 bp deletion in *Ptch1* exon 1. *Disp1*⁻/⁻;*Shh*⁻/⁻;*Ptch1*⁻/⁻;*Ptch2*⁻/⁻ mESCs were additionally heterozygous for 57 bp and 10 bp deletions in *Ptch2* exon 2. *Shh*⁻/⁻ mESCs were heterozygous for 4 bp and 5 bp deletions in *Shh* exon 1. *7dhcr*⁻/⁻; *Shh*⁻/⁻ mESCs were homozygous for a 200 bp deletion encompassing exons 1 and 2 of *7dhcr*.

## Statistics, replicates and confidence intervals

In neural progenitor experiments, technical replicates were independent nEBs. At least 20 nEBs were counted. In Ptch1:LacZ and Gli:Luciferase experiments, technical replicates were independent measurements of a sample lysate. At least three measurements were performed. Outlying technical replicates were never excluded. For all experiments, biological replicates were independent differentiation experiments. Three confirmatory biological replicates were required before reporting, unless stated in figure legend. Biological replicates were disqualified rarely and only when previously characterized and published (>10 confirmatory biological replicates) monotypic nEBs serving as controls (*Smo*⁻/⁻, *Ptch1*^LacZ/LacZ^, *Ptch1*^LacZ/LacZ^;*Ptch2*⁻/⁻) failed to differentiate as expected. Student's t-test was used for all experiments because data was assumed to be normally distributed. Null hypothesis values were obtained by averaging mean and standard deviations of monotypic nEBs and obtaining the inverse of the cumulative normal distribution (N = 10). Median p-values of t-tests between the counts and 10 independent sets of null hypothesis values were reported. This null hypothesis assumes equal contribution of each cell population to nEBs and exclusively cell-autonomous regulation.

## Acknowledgements

*Ptch1*^LacZ/LacZ^ and *Ptch1*^+/LacZ^ mESCs were a gift of Dr. Scott (Stanford University). Smo-/- mESCs were a gift from Dr. Andrew McMahon (University of Southern California). HB9:GFP mESCs were a gift from Dr. Thomas Jessell (Columbia University).This work was supported by NIH grants

R01GM097035 and 1R01GM117090 to HR. BR was a predoctoral fellow of CIRM training grant TG2-01164. We would also like to thank A Luc, B Cole, and J Hardin for their technical assistance.

## Additional information

### Funding

| Funder | Grant reference number | Author |
|---|---|---|
| National Institute of General Medical Sciences | R01GM097035 | Henk Roelink |
| California Institute of Regenerative Medicine | TG2-01164 | Brock Roberts |
| National Institute of General Medical Sciences | 1R01GM117090 | Henk Roelink |

The funders had no role in study design, data collection and interpretation, or the decision to submit the work for publication.

### Author contributions

BR, Performed most experiments, Conception and design, Acquisition of data, Analysis and interpretation of data, Edited the manuscript, Wrote the manuscript, Contributed unpublished essential data or reagents; CC, Performed the *Gli* luciferase experiments, Acquisition of data, Analysis and interpretation of data, Edited the manuscript, Contributed unpublished essential data or reagents; ACA, Generated the Ptch1LacZ/LacZ;Ptch2-/- fibroblasts, Acquisition of data, Analysis and interpretation of data, Edited the manuscript, Contributed unpublished essential data or reagents; CJ, Made the 7dhcr-/- lines and performed experiments with this line, Acquisition of data, Analysis and interpretation of data, Edited the manuscript, Contributed unpublished essential data or reagents; HR, Conception and design, Acquisition of data, Analysis and interpretation of data, Edited the manuscript, Wrote the manuscript

### Author ORCIDs

Henk Roelink, http://orcid.org/0000-0002-5260-3634

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
