## [Decision Letter]

Thank you for resubmitting your work entitled "Patched1 and Patched2 inhibit Smoothened non-cell autonomously" for further consideration at *eLife*. Your revised article has been favorably evaluated by Janet Rossant (Senior editor), a Reviewing editor, and two reviewers.

The manuscript has been improved but there are some remaining issues that need to be addressed before acceptance, as outlined below:

In this revised ms by Roberts et al., the authors test whether the Hh receptors Ptch1 and 2 act non-autonomously to repress Hh signaling in adjacent cells. The authors conclude that a Smoothened inhibitor is secreted by Ptch1/2 expressing cells that acts non-autonomously to inhibit Hh signaling. The mechanism of Ptch inhibition of Hh signaling remains poorly understood and answering this question will be of broad interest. The idea that Ptch might act non-autonomously to inhibit Shh signaling has been around for a some time in the field. Nevertheless the evidence one way or the other has been weak. The additions and alterations the authors have made to their study deal with some of the criticisms that were raised in the initial review. The authors have strengthened their argument by simplifying the flow and presentation of the data and including additional markers in their assays.

Major Comments:

1) The role of Ptch2 in suppressing ligand independent Hh signing in the in vitro assays appears to differ from in vivo, where Ptch2 is thought to have little activity. In the Discussion the authors speculate that Ptch2 levels differ between 'ventral' Ptch1 LacZ/LacZ; Shh-/-, and 'more dorsal' Disp-/-, Ptch1 -/-; Shh-/-, genotypes. This could be tested by qPCR.

2) The authors continue to rely on mixing cells of different genotypes. They provide some evidence that the initial ratios of different cell types are maintained at the later assay time points (Figure 3). However this assay appears to show substantial segregation of some of the genotypes within an EB. Whether some of the cell lines are mixing less well than others might change the interpretation and the authors should address this.

3) Text in the Results implies that it is (Figure 1). This suggests that some Shh signaling is being transduced in these cells and raises questions about the whether these cells can be used to make the epistasis conclusions that the authors propose.

4) Figure 1. Why do Ptch1/2 null fibroblasts have 'low' Shh pathway activity?

5) The authors provide evidence that co-culture of Ptch1 Lacz/Lacz;Ptch2-/-;Shh-/- with Smo-/- cells significantly decreased Nkx2.2+ cells and significantly increased Olig2+ cells. This experiment reintroduces Shh expressing cells into the system (from the Smo-/- cells). This seems to be what the authors are trying to avoid in most experiments in order to simplify interpretation. Smo-/- cells have the effect of reducing Nkx2.2 in neighbouring cells, which is rationalized as being due to the presence of Ptch in these cells, however the effect is 'mild', as Ptch activity itself is reduced by the Shh produced by the Smo-/- cells. This experiment and the convoluted interpretation seems to add little clarity, especially as the comparison is between Ptch1-/-;Ptch2-/-Smo-/- (which have no non-cell autonomous effect on Nkx2.2/Olig2), and Smo-/- (which have slightly reduced Nkx2.2 and increased Olig2).

6) Figure 3: Why are LacZ levels as low (or lower) in Ptch1 LacZ/+ EBS (which produce Shh) as in Ptch1 LacZ/+; Shh-/-? The presence of Shh in Ptch1 LacZ/+ EBS should induce Ptch1 LacZ. Why are LacZ levels lower in Ptch1 LacZ/LacZ; Smo -/- than in Ptch1 LacZ/LacZ; Ptch2-/-; Smo -/-? Smo-/- should abolish all signaling in both cases equivalently.

7) It is unclear how to the DHCR7 data should be interpreted. Why is there a larger inhibition of Hh signaling by 7DHC in cells cocultured with shh-/- or shh-/-;dhcr7-/- cells (Figure 4)? The authors suggest that it could be because Ptch1/2 processes 7DHC into a more potent non-autonomous inhibitor. However it is important to note that deleting DHCR7 or addition of 7DHC to cells will also have substantial feedback effects on cholesterol biosynthetic pathway that could effect multiple cellular processes, notably vesicle trafficking and/or cilia formation. The effects observed in these experiments could therefore be indirect because a change in e.g. vesicle trafficking.

8) The authors will be aware of the recent paper from the Beachy lab- Sever et al. (PNAS) which proposes that instead of 7DHC, 7DHC processed to an oxysterol derivative are the inhibitors. This should be acknowledged

9) Figure 6. The three part mosaics of Ptch1 LacZ/LacZ; Shh-/- or Ptch1 LacZ/LacZ; Ptch2-/-; Shh-/- cells facilitated robust Shh-mediated HB9:GFP+ expression. In contrast, negligible GFP in EBs comprised of Ptch1 LacZ/+; Shh-/- As Ptch1 LacZ/LacZ; Shh-/- cells are unable to inhibit HB9 induction by Shh producing cells this implies that Ptch2 is unable to mediate the same non-autonomous effect as Ptch1. However, this seems inconsistent with Figure 6 where the presence of Ptch2 is sufficient for inhibition, albeit in the absence of Disp. They make an argument about Disp preventing secretion of Ihh and Dhh, however if these ligands were relevant their activity would be evident in the control EBs lacking the 1% Shh producing cells (but still expressing Disp). Figure 6 a very minor fraction of PtcH^+^/LacZ appears able to inhibit HB9+- this suggests considerable range and potency of inhibitory species.

[Editors’ note: a previous version of this study was rejected after peer review, but the authors submitted for reconsideration. The first decision letter after peer review is shown below.]

Thank you for submitting your work entitled "Patched1 and Patched2 inhibit Smoothened non-cell autonomously" for consideration by *eLife*. Your article has been reviewed by three peer reviewers, and the evaluation has been overseen by a Reviewing Editor and Janet Rossant as the Senior Editor. Our decision has been reached after consultation between the reviewers. Based on these discussions and the individual reviews below, we regret to inform you that your work cannot be considered further for publication in *eLife*.

Although all three reviewers found the work of high quality and potentially interesting, the consensus opinion was that there was still a great deal of work required to support the conclusions of the manuscript. Given the time it would take to perform these lengthy additional experiments, we felt it would be most fair to return the paper to you. If in the future, you feel you can address these concerns, we would be open to receiving a new manuscript on the topic and would make every effort to return this to the same reviewers.

Reviewer #1:

In this study Roberts and Roelink use the directed differentiation of wild-type and mutant mouse ES cells to provide evidence that the Hh receptors Ptch1 and 2 act non-autonomously to repress Hh signaling in adjacent cells. They propose that this is due to the secretion of a Smoothened inhibitor by Ptch1/2 expressing cells.

The mechanism of Ptch inhibition of Hh signaling is still poorly understood and the authors are tackling this important question. Although, the idea that Ptch might act non-autonomously to inhibit Shh signaling was proposed several years ago the evidence to support this has been weak and it has remained controversial. The authors' approach to this question is novel and has the potential to provide new insight. However for this study to make a significant contribution to the field several issues need to be addressed.

The first 4 figures of this study establish the reagents the authors then use to test a non-autonomous function of Ptch1/2. While the results appear consistent with the authors' interpretation, the experiments and interpretations need to be strengthened:

The authors rely on mixing cells of different genotypes and assume that the initial ratios of different cell types are maintained at the assay time points. Independent markers (e.g. ubiquitously expressed fluorescent markers) for each of the cell types could be used to demonstrate that the ratios are maintained. This is important because differences in initial set up, growth rates, cell competition or adhesion could be involved. This would change the interpretation of the data.

For example in Figure 7 the authors indicate that they used 1% of cells of a particular genotype. In an EB with c.1000 cells this makes 10 cells with probably considerable variation.

The assays relay either on population, LacZ levels, or the MN marker HB9. LacZ lacks the single cell resolution one would like to see in these assays. HB9 induction is not a direct target of Shh signaling (it is induced in motor neurons that as they differentiate and become post mitotic) and is only expressed in one dorsal-ventral region of the spinal cord. A reduction in HB9 expression could result from either decreased Shh signaling (as the authors conclude) or from increased Shh signaling (if Nkx2.2 expressing progenitors are induced) or from decreased neuronal differentiation (if e.g. pro neural genes are inhibited). Use of a greater range of markers and single cell resolution would greatly strengthen the argument.

The authors argue from the mosaic experiments in Figure 7C,D that the inhibitory activity of Ptch1/2 on adjacent cells is not due to an unknown downstream factor produced by these cells. For this argument they rely on Ptch1-/-;Ptch2-/-;Smo-/- mESCs. However these cells might not be completely inert. The authors state that Ptch1LacZ/LacZ;Ptch2-/-;Smo-/- NEBs expressed Pax7 (p7). However the data in Figure 1 seems to indicate that the amount of Pax7 expression is substantially reduced compared to the other genotypes. This suggests that some Shh signaling is being transduced in these cells and raises questions about the strict epistasis the authors refer to. It also weakens the case for ruling out an unknown factor.

More generally, a limitation of the study is the use of a single experimental design strategy – uncontrolled aggregation of unmarked ESCs of different genotypes. Use of alternative approaches or refining this assay approach is probably required to rigorously test the authors' model. For example:

Marking cells of different genotypes and assembling aggregates with controlled arrangements of different genotypes would remove much of the ambiguity from the data.

Clarifying if the secretion of the inhibitory molecule is the only function of Ptch or if cell-autonomous repression of the Hh pathway involves an independent action of Ptch would be an important insight.

Addressing whether the non-autonomous inhibitory activity is secreted or whether cell contact is necessary would also be an important piece of information. For this type of experiment, the use of transwell assays, or similar, could be used.

Since Ptch molecules have been shown to restrict the spread of Hh ligands in some tissues, the increased Hh response in Ptch double mutant cells could reflect a better spread of Hh ligands through an EB. The authors need to more clearly rule this out.

Reviewer #2:

In the manuscript by Roberts and Roelink, the authors address one of the enduring mysteries of Hh signal transduction, how does Ptch regulate Smo? The authors make two observations. First, through a complex series of experiments using numerous embryonic stem cells lines that lack key components of the Hh pathway, the authors demonstrate that Ptch2 (in the absence of Ptch1) is capable of inhibiting Smo. In a second series of experiments, through co-culture methods, the authors show that the Ptch1 and Ptch2 are capable of inhibiting Smo via non-cell autonomous interactions. From these studies, the authors conclude that Ptch1 and Ptch2 mediate the secretion of a factor that inhibits Smo activation in nearby cells.

Many of the initial experiments support an earlier paper published by the lab (Alfaro et al., 2014). While the experiments presented in this paper were more extensive and genetically clean than those published earlier, the results closely echo conclusions made in the earlier study. The results from the co-culture experiments are novel, interesting, and have important implications for understanding Ptch1-Smo regulation and Hh signaling in tissues. Through multiple experiments, the authors make a case for the existence of a secreted factor mediated by Ptch1/2 that is capable of inhibiting Smo activity. While no factor is directly identified, its existence is intriguing and would be a significant contribution to the large body of work that has been dedicated towards understanding the interaction between Ptch and Smo. While I have some concerns regarding experimental controls (please refer to the Major Issues below), if these can be addressed then I would recommend this manuscript for publication.

Major Issues:

1) In multiple tissue types, Hh signaling has been shown to play a critical role in progenitor cell maintenance. Thus, one could argue that the shifts observed in Figure 1 could be due to changes in progenitor abundance and not due to changes in progenitor identity. While it may be anecdotal, this is actually supported by Figure 1—figure supplement 1, in which the control NEB is by far the largest and is made up of more ventral progenitors than the PtchLacZ/LacZ; Ptch2-/-, Smo-/- NEB. For this reason, I would encourage that a few control experiments be done to determine if there are any differences in (i) progenitor abundance, (ii) neuronal abundance, and (iii) cell death across the various mESC cell lines. If NEB tissue still exists, this should be fairly easy to accomplish. One could use markers such as Caspase3 to assess for cell death, *Sox2* (or any broadly expressed progenitor marker) to assess for progenitor abundance, and Tuj1 or Neun (or any broadly expressed neuronal markers) to assess for neuronal abundance. If there is variability in any of these three factors, the data needs to be normalized to account for these differences.

2) In many of the LacZ measurements, there is no normalization for total viable cell number from which the measurement is being done. In its absence one cannot determine whether a change in LacZ activity is due to changes in cell number or changes in the activity of the reporter. This issue is especially important for Figure 5-how do we know that the effects are not caused by decreases in reporter cell number. Part of the problem is that LacZ is being used as a bulk reporter, rather than a single cell reporter. It would be optimal if the authors could use LacZ measurements in single cells (e.g. by immunostaining in sectioned, chimeric NEBs or FACS) to show non-cell autonomous effects.

3) Given the variability of the NEB size, I would recommend the authors represent the counts in Figure 1, Figure 4, Figure 6, and 7 not as a number of cells per NEB, but rather as a% of cells within the NEB.

4) In the triple chimera experiment shown in Figure 7, it seems to me that an equally likely explanation for the authors' result is that Ptch expressing cells sequester the Shh ligand being secreted by the cells that are producing Shh→ thus leading to reduced HB9-GFP expression in the reporter cells. There is no need to invoke a cell non-autonomous ligand. The authors' should clarify the discussion here and explain more clearly why they do not favor the sequestration model.

Reviewer #3:

This work involves a heroic effort that produced and analyzed an impressive set of mutant mouse ESCs that inactivate components of Shh signal transduction in various combinations. The principal finding – that cells that transduce the Shh signal also down-regulate the Shh response of nearby cells – is supported by the presented evidence, and may represent a new feature that sculpts the landscape of responses signaling proteins. It has important general implications. I recommend publication but strongly recommend a major re-write. The text suffers throughout, and especially in the Discussion, from the trap of second order speculation (speculating on the speculations), and from descriptions and figures that are difficult to follow. Although the imaginative scope is laudable, the extent to which the authors build a hypothetical construct is not.

Samples to illustrate:

Abstract sentence #2: "Patched is a putative proton-driven antiporter, and this antiporter activity is required for Smoothened inhibition." [Patched may well have antiporter activity, but such an activity has never been measured and it is not known if this putative activity is related to inhibition of Smoothened.]

"We found that the Hh response was invariably repressed by the presence of Ptch1/2 in nearby cells. [Was it repressed or not? What does "invariably" add or mean in this context.] We attribute these findings to Ptch1/2-mediated secretion of a Smo inhibitor that affects Smo activity in nearby cells." [There is evidence for inhibition but no evidence for a molecular inhibitor or for secretion. Best not to use genetics to do biochemistry. If the authors want to speculate, it is important that they propose alternative possibilities and indicate which one(s) they favor. As present, the authors build an elaborate scheme by combining assumptions about how Patched works and about a hypothetical inhibitor; they would be better served by emphasizing what their experiments establish without fantasizing molecular models.

---

## [Author Response]

*Major Comments:*

*1) The role of Ptch2 in suppressing ligand independent Hh signing in the* in vitro *assays appears to differ from* in vivo*, where Ptch2 is thought to have little activity. In the Discussion the authors speculate that Ptch2 levels differ between 'ventral' Ptch1 LacZ/LacZ; Shh-/-, and 'more dorsal' Disp-/-, Ptch1 -/-; Shh-/-, genotypes. This could be tested by qPCR.*

This is a valid point that we have discussed in the lab a lot. Before addressing the role of Ptch2, I would like to emphasize that the central conclusion of this paper is based on the *Ptch1*^-/-^;*Ptch2*^-/-^ phenotype, and that this is conserved between the various “families” of cell lines that we have derived. We acknowledge the reality that Ptch1 and Ptch2 activity level are likely subject to selection if either paralog is lost. Our solution to this likelihood has been to generate cell lines genetically null for both paralogs. We followed the reviewers’ suggestion and performed qPCR for *Ptch2* in *Ptch1*^-/-^;*Shh*^-/-^ and *Disp1*^-/-^;*Ptch1*^-/-^;*Shh*^-/-^mESC lines, whose nEBs have differing degrees of ventral identity. *Ptch2* levels are elevated in both lines, relative to wild type cells, and this is specific to differentiating cells, as no differences were detected between cell lines grown in pluripotent culture. *Ptch2* was decreased in *Ptch1*^-/-^;*Shh*^-/-^ nEBs, relative to *Disp1*^-/-^;*Ptch1*^-/-^;*Shh*^-/-^ nEBs, in agreement with our model of Ptch2 as a cell non-autonomous negative regulator of the Shh pathway. This effect was not quite statistically significant (*t-test,* n=3) but trended toward support for our model. We note with interest the relatively large effect that small differences in *Ptch2* expression may have on nEB identity in the absence of Ptch1 (see also comment 9). Nevertheless, we would like to stress the importance of experiments in which cells are genetically null for both
*Ptch1* and *Ptch2,* and thus not subject to variation in *Ptch2* levels.

Author response image 1.**DOI:**
http://dx.doi.org/10.7554/eLife.17634.011

*2) The authors continue to rely on mixing cells of different genotypes. They provide some evidence that the initial ratios of different cell types are maintained at the later assay time points (Figure 3). However this assay appears to show substantial segregation of some of the genotypes within an EB. Whether some of the cell lines are mixing less well than others might change the interpretation and the authors should address this.*

Using the cell tracker dyes, we always find that the initial mixing of mESC cells with distinct genotypes is “pepper and salt”. On occasion, and observed more frequently when the constituent cells acquire different identities, we find that the cell sort in with “like” neighbors. There are two important considerations. 1) Given that we find non cell autonomous effects even in our LacZ assays at early time points, at least some of the critical interactions must take place soon after the nEBs have formed, and are still random in regard to their mosaic distribution. 2) The observed “sorting” of the cells would decrease the number of contacts, and decrease the proximity, between cells with different genotypes and, therefore, suppress the non-autonomous effects we observe in the mosaic nEBs. Sorting alone cannot explain the non-cell autonomous effects we observe, and thus does not affect our conclusions. If anything, the sorting will mask the full extent of the non-autonomous interactions between cell with distinct genotypes.

*3) Text in the Results implies that it is (Figure 1). This suggests that some Shh signaling is being transduced in these cells and raises questions about the whether these cells can be used to make the epistasis conclusions that the authors propose.*

It would seem like that part of this comment was lost, but we assume that this comment is about some level of Hh pathway activation in *Ptch1*^LacZ/LacZ^;*Ptch2*^-/-^;*Smo*^-/-^ cells. This could be concluded from the fewer Pax7 cells seen after differentiation, but perhaps even more so from the higher level of Ptch1:LacZ as shown in Figure 3. These cells have been derived from Matthew Scott’s Ptch1^LacZ/LacZ^mESCs and have undergone some evolution in vitro, reflected in a somewhat less dorsal identity after *Smo* removal. Importantly, under all conditions where we accept/reject the null hypothesis (cells within a mosaic do not affect each other) we take into account levels of LacZ or ventral cells present in the non-mosaic conditions

*4) Figure 1. Why do Ptch1/2 null fibroblasts have 'low' Shh pathway activity?*

We were struck by this observation as well, and we have submitted a paper that studies the Hh response in these cells in depth (Casillas et al., Science Signaling, under review). Importantly, we demonstrate that in *Ptch1*^LacZ/LacZ^;*Ptch2*^-/-^ ES cells the Hh response is not upregulated under normal culture conditions, but only becomes active a few days after serum withdrawal (Figure 3). We hypothesize that the *Ptch1*^LacZ/LacZ^;*Ptch2*^-/-^ fibroblasts have inherited this low level of pathway activation we observe in cultured ES cells lacking *Ptch1* and *Ptch2*. This observation demonstrates that, even in the absence of all Ptch1/2 activity, Smo is not necessarily active. In the submitted paper (Casillas et al.) we show that the *Ptch1*^LacZ/LacZ^;*Ptch2*^-/-^ fibroblasts are insensitive to the Smo inhibitor vismodegib, but that the Hh response can be activated after *Shh* transfection into these cells requiring the N-terminal Cysteine-rich domain (CRD), providing a mechanism of Smo activation distinct from the loss of Ptch1/2 inhibition. We feel these studies illustrate that mechanistic insights remain to be discovered even though at first glance they seem to be at odds with canonical models for Hh signaling. Of note is a recent Nature paper (Byrne et al., July 26 2016) that demonstrates a cholesterol binding pocket in the N-terminal extracellular Cysteine-rich domain (CRD), that is required for Hh signal transduction, further suggesting orthosteric regulation of Smo activity via the CRD.

*5) The authors provide evidence that co-culture of Ptch1 Lacz/Lacz;Ptch2-/-;Shh-/- with Smo-/- cells significantly decreased Nkx2.2+ cells and significantly increased Olig2+ cells. This experiment reintroduces Shh expressing cells into the system (from the Smo-/- cells). This seems to be what the authors are trying to avoid in most experiments in order to simplify interpretation. Smo-/- cells have the effect of reducing Nkx2.2 in neighbouring cells, which is rationalized as being due to the presence of Ptch in these cells, however the effect is 'mild', as Ptch activity itself is reduced by the Shh produced by the Smo-/- cells. This experiment and the convoluted interpretation seems to add little clarity, especially as the comparison is between Ptch1-/-;Ptch2-/-Smo-/- (which have no non-cell autonomous effect on Nkx2.2/Olig2), and Smo-/- (which have slightly reduced Nkx2.2 and increased Olig2).*

We agree that the removal of *Smo* from the “driver cells” does not add new insights into the non-autonomous inhibition. We do think that this experiment has significant confirmatory value in that it lends support for its accompanying mosaic experiment using different cell lines, making it an independent test of our hypothesis. Because the cells are null for *Smo*, we interpret ventrally fated cells in the mosaic nEBs to be lineage restricted to the “readout” cells. We agree that in most experiments, absence of Shh from the system is a major advantage. In this case we reason that because *Ptch1*^-/-^;*Ptch2*^-/-^ “readout” cells cannot respond to ShhN (Figure 1), the experiment has utility. We also reason that Shh in the mosaic nEB should have no effect on *Smo*^-/-^ cells due to the Hh pathway’s quiescence. We have assembled a new supplemental Figure (Figure 2—figure supplement 1) that shows a diagram explaining that the loss of Nkx2.2 in conjunction with a gain of Olig2 is indicative of the acquisition of a more dorsal fate, albeit in subtle fashion.

*6) Figure 3: Why are LacZ levels as low (or lower) in Ptch1 LacZ/+ EBS (which produce Shh) as in Ptch1 LacZ/+; Shh-/-? The presence of Shh in Ptch1 LacZ/+ EBS should induce Ptch1 LacZ. Why are LacZ levels lower in Ptch1 LacZ/LacZ; Smo -/- than in Ptch1 LacZ/LacZ; Ptch2-/-; Smo -/-? Smo-/- should abolish all signaling in both cases equivalently.*

We have no good explanation why under those conditions where we do not detect a measurable Hh response after differentiation (Figure 1) we do get variable levels of LacZ. Importantly, *Ptch1* promotor activity (which is measured with the LacZ) is not exclusively on the control of Smo activity, and low levels might vary due to reasons unknown to us. The contribution of the *Shh* locus remains minimal. The loss of Shh causes a slight increase in the number of Pax7 positive cells as compared to the equivalent cell line with the Shh locus intact (Figure 1AC), while the loss of Shh causes a small increase in Ptch1 promotor activity (Figure 3). In both cases we consider the Hh response to be effectively off, despite the variable low levels of LacZ.

*7) It is unclear how to the DHCR7 data should be interpreted. Why is there a larger inhibition of Hh signaling by 7DHC in cells cocultured with shh-/- or shh-/-;dhcr7-/- cells (Figure 4)? The authors suggest that it could be because Ptch1/2 processes 7DHC into a more potent non-autonomous inhibitor. However it is important to note that deleting DHCR7 or addition of 7DHC to cells will also have substantial feedback effects on cholesterol biosynthetic pathway that could effect multiple cellular processes, notably vesicle trafficking and/or cilia formation. The effects observed in these experiments could therefore be indirect because a change in e.g. vesicle trafficking.*

It is important to note that we measure the Hh response only in cells that are genetically intact for DHCR7, and are thus “normal” in their cholesterol synthesis pathway. We agree that the loss of *dhcr7* can result in significant changes in a cell, but it is reasonable to assume that effects on vesicle trafficking and/or cilia formation are cell autonomous, and thus restricted to a mosaic compartment in which we do not measure the Hh response. It is indeed remarkable that altered cholesterol biosynthesis, can affect the Hh response in cells with a normal cholesterol biosynthesis, but lacking *Ptch1/2*.

The addition of 7DHC is indeed more complex, since both compartments in the mosaic are equally exposed. It is true that 7DHC inhibits the Hh response in cells lacking *Ptch1/2*, but our mosaic experiments demonstrate that the inclusion of *Ptch1/2* containing cells exacerbates the inhibitory effect of 7DHC on cells that lack *Ptch1/2*. We hypothesize that this non autonomous effect is mediated by Ptch1/2 activity. We have added additional clarifying interpretive language to the text.

*8) The authors will be aware of the recent paper from the Beachy lab- Sever et al. (PNAS) which proposes that instead of 7DHC, 7DHC processed to an oxysterol derivative are the inhibitors. This should be acknowledged*

This paper came out the same day we submitted, and we have added references to this relevant paper in the revised manuscript.

*9) Figure 6. The three part mosaics of Ptch1 LacZ/LacZ; Shh-/- or Ptch1 LacZ/LacZ; Ptch2-/-; Shh-/- cells facilitated robust Shh-mediated HB9:GFP+ expression. In contrast, negligible GFP in EBs comprised of Ptch1 LacZ/+; Shh-/- As Ptch1 LacZ/LacZ; Shh-/- cells are unable to inhibit HB9 induction by Shh producing cells this implies that Ptch2 is unable to mediate the same non-autonomous effect as Ptch1. However, this seems inconsistent with Figure 6 where the presence of Ptch2 is sufficient for inhibition, albeit in the absence of Disp. They make an argument about Disp preventing secretion of Ihh and Dhh, however if these ligands were relevant their activity would be evident in the control EBs lacking the 1% Shh producing cells (but still expressing Disp). Figure 6 a very minor fraction of PtcH^+^/LacZ appears able to inhibit HB9+- this suggests considerable range and potency of inhibitory species.*

We reached similar conclusions and agree with these comments. We suspected that differences in GFP induction in these two families of mESC lines might stem from differences in Ptch2 expression. Our qPCR data submitted with this revision supports this view (see comment 1). We agree that the range and potency of the inhibitory species has great relevance to elucidating the Hh signaling mechanism. We agree that contributions from Ihh and Dhh are unlikely, but it is still formally possible that control nEBs lacking Shh expressing cells are marginally activated by Dhh/Ihh at low levels insufficient to activate HB9:GFP. The use of *Disp^1^*^-/-^mESCs serves as an additional safeguard against this possibility, as well as an independent hypothesis test.

[Editors’ note: the author responses to the first round of peer review follow.]

*Reviewer #1:*

*The authors rely on mixing cells of different genotypes and assume that the initial ratios of different cell types are maintained at the assay time points. Independent markers (e.g. ubiquitously expressed fluorescent markers) for each of the cell types could be used to demonstrate that the ratios are maintained. This is important because differences in initial set up, growth rates, cell competition or adhesion could be involved. This would change the interpretation of the data.*

We used vital cell markers (Cell Tracker dyes) to determine that cells contributed to mosaic nEBs in the predicted ratios and demonstrate this data for the most quantitative assays used.

The mosaic contribution in nEBs did not differ from predicted. Dye signal persisted reliably for 60h in mosaic nEBs, and we limited their use to the shorter (LacZ) experiments. We were unable to generate a truly ubiquitously expressed fluorescent marker as mESCs silenced at some rate all constructs using a ubiquitous promoter, and thus we could only rely on the vital dyes.

*For example in Figure 7 the authors indicate that they used 1% of cells of a particular genotype. In an EB with c.1000 cells this makes 10 cells with probably considerable variation.*

*The assays relay either on population, LacZ levels, or the MN marker HB9. LacZ lacks the single cell resolution one would like to see in these assays. HB9 induction is not a direct target of Shh signaling (it is induced in motor neurons that as they differentiate and become post mitotic) and is only expressed in one dorsal-ventral region of the spinal cord. A reduction in HB9 expression could result from either decreased Shh signaling (as the authors conclude) or from increased Shh signaling (if Nkx2.2 expressing progenitors are induced) or from decreased neuronal differentiation (if e.g. pro neural genes are inhibited). Use of a greater range of markers and single cell resolution would greatly strengthen the argument.*

We maintained use of Ptch1:LacZ because we can very accurately measure it in aggregate, which we argue is more crucial than single cell resolution. We found Ptch1:LacZ staining to be unreliable and ambiguous due to the need to integrate the frequency of positive cells and staining intensity.

We do not detect variation in our nEBs with regard to their acquisition of Pax6+ neural progenitor fate, or Tuj1+ neural fate. Use of a Sim1 reporter line to mark the most ventral neural domain was unsuccessful due to the paucity of these cells, perhaps due to their need for high signaling.

We demonstrate that HB9:GFP+ cells colabel with Isl1/2 staining, which we have found to correlate positively with Smo activation in nEBs. We also demonstrate that HB9:GFP has a positive relationship with Smo activity, as demonstrated by a range of SAG doses. Furthermore, the quantification is routinely base on the analysis of >20 nEBs, making random variation much less of a concern.

*The authors argue from the mosaic experiments in Figure 7C,D that the inhibitory activity of Ptch1/2 on adjacent cells is not due to an unknown downstream factor produced by these cells. For this argument they rely on Ptch1-/-;Ptch2-/-;Smo-/- mESCs. However these cells might not be completely inert. The authors state that Ptch1LacZ/LacZ;Ptch2-/-;Smo-/- NEBs expressed Pax7 (p7). However the data in Figure 1 seems to indicate that the amount of Pax7 expression is substantially reduced compared to the other genotypes. This suggests that some Shh signaling is being transduced in these cells and raises questions about the strict epistasis the authors refer to. It also weakens the case for ruling out an unknown factor.*

We regret the confusion caused by our original figure and have amended it. Ptch1LacZ/LacZ;Ptch2- /-;Smo-/- mESCs reliable give rise Pax7+ cells in nEBs. Pax7+ cells arrange in distinct domains in nEBs and are rarely distributed throughout the tissue. Ptch1LacZ/LacZ;Ptch2-/-;Smo-/- cells generate smaller nEBs when cultured monotypically and it is somewhat more common to obtain nEBs devoid of Pax7+ cells in a given culture, but many nEBs are Pax7+. We demonstrate box and whisker plots for transparency with regard to this reality.

*More generally, a limitation of the study is the use of a single experimental design strategy – uncontrolled aggregation of unmarked ESCs of different genotypes. Use of alternative approaches or refining this assay approach is probably required to rigorously test the authors' model. For example:*

*Marking cells of different genotypes and assembling aggregates with controlled arrangements of different genotypes would remove much of the ambiguity from the data.*

We use Cell Tracker markers to address this criticism and we also employ our panel of cell lines in various combinations in order to offset these limitations and test the robustness of our assay.

*Clarifying if the secretion of the inhibitory molecule is the only function of Ptch or if cell-autonomous repression of the Hh pathway involves an independent action of Ptch would be an important insight.*

We agree with this but have elected to focus our experiments on establishing the veracity of these effects in this system, with an ultimate ambition to more mechanistically interrogate it with more extensive gene editing. We view these planned experiments as beyond the scope of this study. Current approaches restrict us to expressing Ptch1/2 variants in Ptch1/2 null cells to test these mechanistic subtleties. We elect to limit our conclusions at this time to Ptch1/2 effects when expressed endogenously. We will use gene editing to generate mutations in cell lines in the future.

*Addressing whether the non-autonomous inhibitory activity is secreted or whether cell contact is necessary would also be an important piece of information. For this type of experiment, the use of transwell assays, or similar, could be used.*

We agree with this notion. However, we elect to use the privileged environment within the nEB because of its heightened physiological relevance. We are dubious as to whether Ptch1/2 expressed at levels meaningful for a realistic induction response can condition medium as has been reported in fibroblasts, for example. Nor do we value that observation highly given the reality of signaling in the embryo, where cells are in close contact. Our experiments to date using medium conditioned by mESCs with or without Ptch1/2 failed to show Smo modulating activity.

*Since Ptch molecules have been shown to restrict the spread of Hh ligands in some tissues, the increased Hh response in Ptch double mutant cells could reflect a better spread of Hh ligands through an EB. The authors need to more clearly rule this out.*

We demonstrate using live staining with 5E1 anti-Shh that Shh distribution within cell lines in our panel does not meaningfully differ.

Reviewer #2:

*Major Issues:*

*1) In multiple tissue types, Hh signaling has been shown to play a critical role in progenitor cell maintenance. Thus, one could argue that the shifts observed in Figure 1 could be due to changes in progenitor abundance and not due to changes in progenitor identity. While it may be anecdotal, this is actually supported by Figure 1—figure supplement 1, in which the control NEB is by far the largest and is made up of more ventral progenitors than the PtchLacZ/LacZ; Ptch2-/-, Smo-/- NEB. For this reason, I would encourage that a few control experiments be done to determine if there are any differences in (i) progenitor abundance, (ii) neuronal abundance, and (iii) cell death across the various mESC cell lines. If NEB tissue still exists, this should be fairly easy to accomplish. One could use markers such as Caspase3 to assess for cell death, Sox2 (or any broadly expressed progenitor marker) to assess for progenitor abundance, and Tuj1 or Neun (or any broadly expressed neuronal markers) to assess for neuronal abundance. If there is variability in any of these three factors, the data needs to be normalized to account for these differences.*

We find that Pax6+ cells and Tuj+ cells exist in roughly equal frequencies in nEBs derived exclusively from each cell line and we demonstrate this now. *Sox2* expression also does not meaningfully differ although we do not show this data.

*2) In many of the LacZ measurements, there is no normalization for total viable cell number from which the measurement is being done. In its absence one cannot determine whether a change in LacZ activity is due to changes in cell number or changes in the activity of the reporter. This issue is especially important for Figure 5-how do we know that the effects are not caused by decreases in reporter cell number. Part of the problem is that LacZ is being used as a bulk reporter, rather than a single cell reporter. It would be optimal if the authors could use LacZ measurements in single cells (e.g. by immunostaining in sectioned, chimeric NEBs or FACS) to show non-cell autonomous effects.*

We address this concern by tracking contributions from cell lines in mosaic nEBs with vital dye markers, and confirm that they are as expected, and we demonstrate this data.

*3) Given the variability of the NEB size, I would recommend the authors represent the counts in Figure 1, Figure 4, Figure 6, and 7 not as a number of cells per NEB, but rather as a% of cells within the NEB.*

We have performed a similar analysis where nEBs are normalized for their size and have found that it does not alter our conclusions but that the effects we report are actually more pronounced. Because we did not feel comfortable with the assumption that cell fate would be independent of nEB size, we chose to report the most conservative conclusion.

*4) In the triple chimera experiment shown in Figure 7, it seems to me that an equally likely explanation for the authors' result is that Ptch expressing cells sequester the Shh ligand being secreted by the cells that are producing Shh→ thus leading to reduced HB9-GFP expression in the reporter cells. There is no need to invoke a cell non-autonomous ligand. The authors' should clarify the Discussion here and explain more clearly why they do not favor the sequestration model.*

We address this concern with Shh staining, which we demonstrate is equivalent in mosaic nEBs comprised of each cell line in our panel. We have modified the Discussion making our argument that Shh sequestration (or the lack thereof) is not the main reason for the effects observed.

In short: 1) All relevant lines are Shh-/-, and thus not subject to sequestration, 2) Enhancing the SAG effects argues that the Ptch1/2 cargo is acting on Smo, and 3) no alteration of extracellular Shh distribution as a function of Ptch1/2 in the surrounding cells is observed.

*Reviewer #3:*

*This work involves a heroic effort that produced and analyzed an impressive set of mutant mouse ESCs that inactivate components of Shh signal transduction in various combinations. The principal finding – that cells that transduce the Shh signal also down-regulate the Shh response of nearby cells – is supported by the presented evidence, and may represent a new feature that sculpts the landscape of responses signaling proteins. It has important general implications. I recommend publication but strongly recommend a major re-write. The text suffers throughout, and especially in the Discussion, from the trap of second order speculation (speculating on the speculations), and from descriptions and figures that are difficult to follow. Although the imaginative scope is laudable, the extent to which the authors build a hypothetical construct is not.*

We regret the confusion our original manuscript caused and have taken extensive efforts to rewrite it with greater clarity.

*Samples to illustrate:*

*Abstract sentence #2: "Patched is a putative proton-driven antiporter, and this antiporter activity is required for Smoothened inhibition." [Patched may well have antiporter activity, but such an activity has never been measured and it is not known if this putative activity is related to inhibition of Smoothened.]*

*"We found that the Hh response was invariably repressed by the presence of Ptch1/2 in nearby cells. [Was it repressed or not? What does "invariably" add or mean in this context.] We attribute these findings to Ptch1/2-mediated secretion of a Smo inhibitor that affects Smo activity in nearby cells." [There is evidence for inhibition but no evidence for a molecular inhibitor or for secretion. Best not to use genetics to do biochemistry. If the authors want to speculate, it is important that they propose alternative possibilities and indicate which one(s) they favor. As present, the authors build an elaborate scheme by combining assumptions about how Patched works and about a hypothetical inhibitor; they would be better served by emphasizing what their experiments establish without fantasizing molecular models.*

We have greatly revised the degree to which we draw conclusions based on our experiments and have confined our speculations.